# SUBEQUIVARIANT MORPHOLOGY-BEHAVIOR CO-EVOLUTION IN 3D ENVIRONMENTS

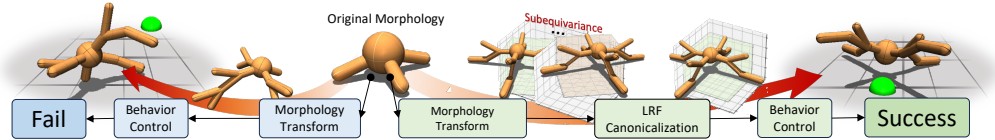

Figure 1: The left illustrates uncoordinated morphology and control without subequivariance, resulting in task failure. The right demonstrates symmetry in morphology and control through subequivariant co-evolution, enabling successful goal completion.

## ABSTRACT

The co-evolution of morphology and behavior in 3D space has garnered considerable interest in the field of embodied intelligence. While recent studies have highlighted the considerable benefits of geometric symmetry for tasks like learning to locomote, navigate, and explore in dynamic 3D environments, its role within co-evolution setup remains unexplored. Existing benchmarks encounter several key issues: 1) the task lacks consideration for spatial geometric information; 2) the method lacks geometric symmetry to deal with the complexities in 3D environments. In this work, we propose a novel setup, named Subequivariant Morphology-Behavior Co-Evolution in 3D Environments (3DS-MB), to address the identified limitations. To be specific, we propose EquiEvo, which injects geometric symmetry, i.e., subequivariance, to construct dynamic, learnable local reference frames, enabling the joint policy to generalize to diverse task spatial structures, thereby improving co-evolution efficiency. Then, we evaluate EquiEvo on the proposed environments, where our method consistently and significantly outperforms existing approaches in tasks such as locomotion navigation and adversarial scenarios. Extensive experiments underscore the importance of subequivariance for the co-evolution of morphology and behavior, effective morphology-task mapping and robust morphology-behavior mapping.

## 1    INTRODUCTION

The rich diversity of animal morphologies, shaped over millions of years in complex environments, underscores the deep link between body form and intelligence, where well-adapted morphologies enable agents to learn and perform complex tasks effectively (Gupta et al., 2021; Brooks, 1991). Morphology evolution is fundamentally driven by environmental interactions and task demands, as supported by evolutionary biology (Maynard-Smith, 1974; Gould, 2010). The co-evolution of morphology and behavior in 3D environments has gained significant interest (Sims, 1994; Dong et al., 2023; Chen et al., 2023b; Gupta et al., 2021), focusing on how agents' form and function evolve together to improve adaptability and performance. Some works, such as (Gupta et al., 2021; Dong et al., 2023), impose bilateral symmetry constraints directly on morphology, diverging from evolutionary principles, as agents should evolve optimal morphologies through interaction with the environment, rather than relying on predefined constraints.

Alternatively, the geometric symmetry of environments and dynamic systems naturally exists, which motivates us to explore how leveraging such symmetry can accelerate the evolution of optimal morphologies through interaction with the environment. Recent studies highlight that leveraging geometric symmetry in tasks like locomotion, navigation, and exploration within dynamic 3D settings

can significantly improve the efficiency of behavioral evolution (Chen et al., 2023a; 2024). While recent co-evolution benchmarks (Yuan et al., 2022; Huang et al., 2024; Liu et al., 2022) have achieved remarkable progress (Liu et al., 2022), they often fail to fully exploit geometric symmetry within 3D spaces. These benchmarks face two main issues: 1) a focus on tasks with limited spatial geometric considerations, such as the "move forward" objective, which are simplified to fixed directional movements and do not require complex direction-aware strategies; 2) a lack of research exploring the use of geometric symmetry in co-evolution setups to handle the directional complexities of 3D environments effectively.

In this work, we address the limitations of current co-evolution benchmarks by focusing on tasks that require rich 3D spatial understanding and incorporate dynamic interactions. To achieve this, we propose a novel setup, named Subequivariant Morphology-Behavior Co-Evolution in 3D Environments (3DS-MB). Our setup extends two core tasks to enable co-evolution in complex 3D spatial environments: navigation and sumo (Chen et al., 2023a; Huang et al., 2024). In the navigation task, the agent moves towards randomly generated goals, requiring continuous adaptation to new spatial contexts. In the sumo task, two agents compete in an arena, where the objective is to push the opponent out. The constantly changing adversarial dynamics require agents to adapt to varying attack and defense angles, demanding both strategic behavior and morphology adaptation to effectively leverage rich directional information. These tasks introduce variability in movement and environmental interaction, challenging agents to develop robust, generalized policies that account for diverse geometric conditions.

To effectively handle the directional complexities of 3D environments, we incorporate subequivariant neural networks within our co-evolution framework. As illustrated in Figure 1, we introduce subequivariant graph neural networks (Chen et al., 2023a; Han et al., 2022a) to predict a dynamic, adaptive Local Reference Frame (LRF) for the agent (Chen et al., 2024). By projecting directional vectors from the global world frame into this LRF, we transform spatial representations into an invariant form, preserving geometric symmetry and simplifying the search space. For instance, under rotational symmetry, states and actions in any direction can be treated as equivalent, effectively reducing a 2D/3D problem to a simpler 1D/2D one. This approach enables the policy to generalize across diverse spatial structures, improving both generalization ability and sample efficiency in the co-evolution process, allowing agents to efficiently learn and adapt to the complex directional information inherent in 3D tasks. An essential aspect of our approach is ensuring that the evolution of morphology remains invariant to geometric transformations, such as rotations or translations, as these do not alter the intrinsic properties of the environment.

Our contributions can be summarized as follows:

- We introduce 3DS-MB, a benchmark for subequivariant morphology-behavior co-evolution, designed to capture the rich spatial geometric information and variability inherent in complex 3D environments.

- We propose EquiEvo, a novel co-evolution framework that leverages subequivariant graph neural networks to effectively incorporate geometric symmetry, enhancing both morphology adaptation and behavior control.

- We empirically validate the performance of EquiEvo on the 3DS-MB, where our approach consistently outperforms existing methods. Extensive ablation studies and analyses further demonstrate the framework's robustness and highlight the benefits of integrating subequivariance in co-evolution tasks.

## 2 PRELIMINARIES

**Morphology-Behavior Co-Evolution.** To account for the effect of morphology to agent's behavior, the typical MDP formulation could be extended by incorporating the agent's morphology. Specifically, we define the extended MDP as $\mathcal{M} = (\mathcal{S}, \mathcal{A}, \mathcal{T}, R, \gamma, \mathcal{G}_{\mathrm{m}})$ of state space, action space, transition dynamics, a reward function, a discount factor and the agent's morphology. $\mathcal{G}_{\mathrm{m}}$ is typically represented as a graph. The transition dynamics $P(s_{t+1}|s_t, a_t, \mathcal{G}_{\mathrm{m}})$ and the control policy $\pi(a_t|s_t, \mathcal{G}_{\mathrm{m}})$ are all conditioned on $\mathcal{G}_{\mathrm{m}}$. The expected return depends on both the policy $\pi$ and the morphology $\mathcal{G}_{\mathrm{m}}$: $J(\pi, \mathcal{G}_{\mathrm{m}}) := \mathbb{E}_{\pi, \mathcal{G}_{\mathrm{m}}} [\sum_{t=0}^{\infty} \gamma^t r_t]$. The morphology-behavior co-evolution problem

can be mathematically described as optimizing the following two-layer objective:

$$\mathcal{G}_{\mathrm{m}}^* := \arg\max_{\mathcal{G}_m} J(\pi_{\mathcal{G}}, \mathcal{G}_{\mathrm{m}})$$

$$\text{s.t.} \quad \pi_{\mathcal{G}} = \arg\max_{\pi} J(\pi, \mathcal{G}_{\mathrm{m}}). \tag{1}$$

Building on the classical policy optimization algorithm, PPO (Schulman et al., 2017), Yuan et al. (2022) propose a unified approach to jointly optimize two sub-policies: morphology transformation and behavior control. Specifically, the policy is structured to first transform the agent's morphology and develop its attributes, which is then followed by behavior control for the resulting morphology within each episode.

**Geometric Symmetry.** Learning in 3D environments is hard due to the excess requirement of exploration. To relieve the difficulty of co-evolution in 3D environments with the massive search space, the inherent geometric symmetry of 3D environments should be exploited. Intuitively, if the entire environment undergoes rigid transformations such as rotations and translations, the perceived environment should remain unchanged from the perspective of an agent within it. In other words, the agent should remain unaffected by global geometric transformations. This property is formally captured by the concept of *equivariance*, defined as follows:

**Definition 1** (Equivariance). *Suppose $\vec{z}^1$ to be vector features (positions, velocities, etc) that are steerable by a group $G$, and $\boldsymbol{h}$ non-steerable scale features. A function $f$ is $G$-equivariant, if for any transformation $g \in G, \forall \vec{z} \in \mathbb{R}^{3 \times m}, \boldsymbol{h} \in \mathbb{R}^d, f(g \cdot \vec{z}, \boldsymbol{h}) = g \cdot f(\vec{z}, \boldsymbol{h})$. Similarly, $f$ is invariant if $f(g \cdot \vec{z}, \boldsymbol{h}) = f(\vec{z}, \boldsymbol{h})$. Here $\cdot$ denotes the group operation.*

To adhere to the principles of classical physics under the influence of gravity in typical 3D environments, we consider equivariance over a specific subgroup of $\mathrm{E}(3)^2$ defined as $\mathrm{E}_{\vec{g}}(3)$. This subgroup is composed of $\mathrm{O}_{\vec{g}}(3) := \{\boldsymbol{O} \in \mathbb{R}^{3 \times 3} \mid \boldsymbol{O}^\top \boldsymbol{O} = \boldsymbol{I}, \boldsymbol{O}\vec{g} = \vec{g}\}$ and $\mathrm{T}_{\vec{g}}(3) := \{\vec{t} \in \mathbb{R}^3 \mid \vec{t} \times \vec{g} = \vec{0}\}$. The group operation of $\mathrm{E}_{\vec{g}}(3)$ is instantiated as $g \cdot \vec{z} := \boldsymbol{O}\vec{z} + \vec{t}^3$. In this way, $\mathrm{E}_{\vec{g}}(3)$-equivariance is constrained to translations, rotations, and reflections along the direction of $\vec{g}$. We refer to this property as *subequivariance*, specifically highlighting $\mathrm{E}_{\vec{g}}(3)$-equivariance. This relaxation of group constraints is critical in environments influenced by gravity, as it allows the model to capture gravitational effects. Unless otherwise specified, in the following text, we will use the term "equivariance" to refer to $\mathrm{E}_{\vec{g}}(3)$-equivariance, and "invariance" to refer to $\mathrm{E}_{\vec{g}}(3)$-invariance.

**Subequivariant Graph.** Subequivariant graph is an extention of geometric graph (Han et al., 2024), defined as $\vec{\mathcal{G}}_{\vec{g}} := (\mathcal{V}, \mathcal{E}, \boldsymbol{H}, \vec{\boldsymbol{Z}}, \vec{g})$. $\boldsymbol{H} \in \mathbb{R}^{N \times C_h}$ contains scalar node features of $C_h$ channels. $\vec{\boldsymbol{Z}} \in \mathbb{R}^{N \times 3 \times C_z}$ contains vector node features of $C_z$ channels. $\vec{g}$ indicates that the permitted transformations to the graph must be elements of $\mathrm{E}_{\vec{g}}(3)$. The transformation of subequivariant graph by $g \in \mathrm{E}_{\vec{g}}(3)$ is defined as $g \cdot \vec{\mathcal{G}}_{\vec{g}} := (\mathcal{V}, \mathcal{E}, \boldsymbol{H}, g \cdot \vec{\boldsymbol{Z}}, \vec{g})$. To effectively make use of subequivariant graph, previous works have adopted a universally expressive construction of $\mathrm{O}_{\vec{g}}(3)$-equivariant function (Han et al., 2022a; Chen et al., 2023a; 2024):

$$f_{\vec{g}}(\vec{z}, \boldsymbol{h}) = [\vec{z}, \vec{g}]\sigma([\vec{z}, \vec{g}]^\top [\vec{z}, \vec{g}], \boldsymbol{h}), \tag{2}$$

where $\sigma(\cdot)$ is an Multi-Layer Perceptron and [ ] denotes concatenation along the last dimension.

## 3 SETUP AND METHOD

Our goal is to learn a co-evolution policy in 3D environments that simultaneously adapts the agent's morphology and enhances behavior control skills to optimize task performance. To achieve this, we first introduce our newly proposed 3D morphology-behavior co-evolution setup, termed 3DS-MB,

Table 1: Comparison of Setups.

| Aspect | Prior | 3DS-MB |
|---|---|---|
| Task | Fixed Directions | Variable Directions |
| Graph | Topology Graph | Subequivariant Graph |
| Group | $\emptyset$ | $\mathrm{E}_{\vec{g}}(3)$ |

---

[1] Note that we use a right-arrow superscript on $\vec{z}$ to distinguish it from the scalar $\boldsymbol{h}$, which remains unaffected by the transformations.

[2] The symmetrical structure of 3D environments is captured by $\mathrm{E}(3)$, the 3-dimensional Euclidean group, which includes rotations, reflections, and translations.

[3] Note that $\vec{t}$ only acts on the 3D coordinate vector.

which incorporates tasks involving richer 3D information. We then present our co-evolution framework, EquiEvo (illustrated in Figure 2), which leverages the symmetric structure of 3D environments through subequivariant neural networks to efficiently evolve morphology and learn control skills. We also outline our method in Algorithm 1.

## 3.1 EXTENSION TO 3DS-MB

**Setups.** Exploring how agents evolve and behave in tasks involving complex state information of 3D environments is crucial, as factors like positions, velocities, and directions are inherently represented as vectors.

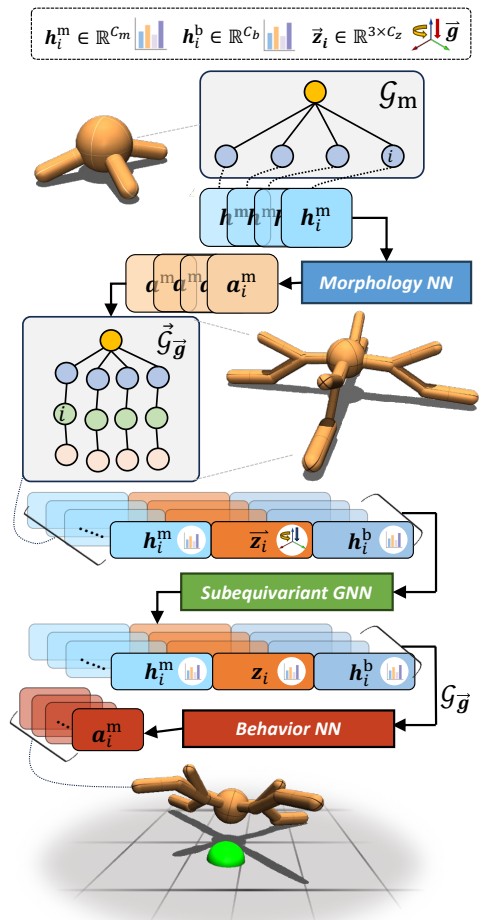

Figure 2: Flowchart of EquiEvo.

While prior works have made significant strides, existing benchmarks (Yuan et al., 2022; Huang et al., 2024) primarily focus on tasks with fixed directions and limited consideration of those requiring richer vector representations, particularly tasks involving variable directions. Consequently, there is no need to utilize symmetry groups, and modeling with topology graphs is sufficient for co-evolution of morphology and behavior in their tasks. In contrast, our proposed 3DS-MB setup introduces tasks with variable directions, employs subequivariant graphs, and leverages the group $\mathrm{E}_{\vec{g}}(3)$ to capture the geometric properties of 3D spaces. A detailed comparison of the environment setups between 3DS-MB and prior benchmarks is provided in Table 1.

**Invariant of Morphology Value.** The evolution of morphology should be invariant to geometric transformations like rotations or translations, as these do not alter the intrinsic properties of the environment. From the perspective of PPO optimization, to optimize the morphology transform policy, we need to estimate the value of morphology transform actions, which must be invariant to $\mathrm{E}_{\vec{g}}(3)$ transformations. For example, consider a scenario where an agent with the same morphology is initialized in a state that has been rotated within the environment. The behavior should rotate correspondingly, ensuring that the morphology value determined by behavior remains consistent. In co-evolution frameworks like Transform2Act (Yuan et al., 2022), morphology transformation and behavior control are treated as different steps within a single episode. Consistent feedback for morphology transformation can only be achieved by leveraging geometric symmetry throughout the interaction with the environment. To achieve this, we integrate equivariant networks into the behavior control policy. By inject symmetry into the model architecture, the output is constrained to be equivariant, provides consistent value for morphology value estimation, and facilitates the co-evolution learning process. Furthermore, equivariant networks have been proven to learn control policies more efficiently, which is crucial for reducing the complexity of solving the complex two-layer optimization problem. The role of geometric symmetry in morphology evolution is discussed in detail in Appendix D.

## 3.2 EQUIEVO

**Agent Representation.** The morphology of the agent is a graph $\mathcal{G}_{\mathrm{m}} := (\mathcal{V}, \mathcal{E}, \boldsymbol{H}^m)$. where each node $i \in \mathcal{V}$ represents a limb, and each edge $(i, j) \in \mathcal{E}$ corresponds to the joint that connects the limbs $i$ and $j$. The morphology structure of the agent is represented by the topology of graph. $\boldsymbol{H}^{\mathrm{m}}$

contains morphology attributes for the agent's limbs, joints, and the specific design for the structure given by the graph. For each node $i$, $\boldsymbol{h}_i^{\mathrm{m}} \in \mathbb{R}^{C_{\mathrm{m}}}$ includes the length and size of the limb, joint torque limit, the node depth in the body tree, etc.

The agent's state is defined as a subequivariant graph $\vec{\mathcal{G}}_{\vec{g}} := (\mathcal{V}, \mathcal{E}, \boldsymbol{H}^{\mathrm{m}}, \boldsymbol{H}^{\mathrm{b}}, \vec{\boldsymbol{Z}}, \vec{g}) = (\mathcal{G}, \boldsymbol{H}^{\mathrm{b}}, \vec{\boldsymbol{Z}}, \vec{g})$. Each node $i$ has scalar state and vector state, expressed as $\boldsymbol{h}_i^{\mathrm{b}} \in \boldsymbol{H}^{\mathrm{b}}$ and $\vec{z}_i \in \vec{\boldsymbol{Z}}$, respectively. The scalar state $\boldsymbol{h}_i^{\mathrm{b}} \in \mathbb{R}^{C_{\mathrm{b}}}$ contains scalars representing the configuration of the joint, including joint angle, joint angular velocity, etc. The vector joint state $\vec{z}_i \in \mathbb{R}^{3 \times C_{\mathrm{m}}}$ contains position $\vec{p}_i \in \mathbb{R}^3$, direction, linear velocity, angular velocity, etc. Here, position $\vec{p}_i$ is transformed into a translation-invariant representation by subtracts position of root node, thereby ensuring translation invariance. Notably, the morphology attributes and the scalar state are invariant, while the vector state is equivariant.

We emphasize the differences between $\mathcal{G}_{\mathrm{m}}$ and $\mathcal{G}_{\vec{g}}$. $\mathcal{G}_{\mathrm{m}}$ solely represents the agent's morphology and does not interact with the environment. In contrast, $\mathcal{G}_{\vec{g}}$ interacts with the environment, involving $\boldsymbol{H}^b$ and $\vec{\boldsymbol{Z}}$, which are obtained during the rollout process.

**Morphology Transform.** In the morphology transform stage, starting from the original morphology, the agent undergoes morphology transform actions $\boldsymbol{a}^{\mathrm{m}}$ to modify the structure and attributes through the morphology transform sub-policy $\pi_\theta^{\mathrm{m}}$. We adopted the two-stage design introduced in Transform2Act (Yuan et al., 2022). The transform actions $\boldsymbol{a}^{\mathrm{m}} \in \{\boldsymbol{a}^{\mathrm{s}}, \boldsymbol{a}^{\mathrm{a}}\}$ include two types: (1) Structure transform actions $\boldsymbol{a}^{\mathrm{s}}$, which are discrete and change the topological structure of the morphology by adding or removing joints and limbs; (2) Attribute transform actions $\boldsymbol{a}^{\mathrm{a}}$, which can be continuous or discrete, used to modify the specific instantiation of the given structure. Accordingly, $\pi_\theta^{\mathrm{m}}$ is a composition of the structure transform sub-policy $\pi_\theta^{\mathrm{s}}$ and the attribute transform policy $\pi_\theta^{\mathrm{a}}$. We divide the morphology transform stage into two sub-stages - structure transform stage and attribute transform stage - and use $\Phi$ as the stage identifier. We can represent the morphology transform sub-policy as follows:

$$\pi_\theta^{\mathrm{m}}(\boldsymbol{a}^{\mathrm{m}}|\mathcal{G}_{\mathrm{m}}, \Phi) := \begin{cases} \pi_\theta^{\mathrm{s}}(\boldsymbol{a}^{\mathrm{s}}|\mathcal{G}_{\mathrm{m}}, \Phi), & \text{if } \Phi = \text{Structure} \\ \pi_\theta^{\mathrm{a}}(\boldsymbol{a}^{\mathrm{a}}|\mathcal{G}_{\mathrm{m}}, \Phi), & \text{if } \Phi = \text{Attribute} \end{cases} \tag{3}$$

The structural transform stage lasts for $N_{\mathrm{s}}$ steps, while the attribute transform stage spans $N_{\mathrm{a}}$ steps. The resulting morphology is then utilized by the behavior sub-policy to interact with the environment. Notably, during the morphology transform stage, no environment reward is assigned to the agent, as there is no direct interaction with the environment. Instead, the morphology transform sub-policy is updated solely based on feedback from the subsequent behavior control stage.

**Local Reference Frame Canonicalization.** According to Han et al. (2024), the Local Reference Frame (LRF) Canonicalization is a common technique in geometric graph neural networks to achieve equivariance (or invariance). We chose to use the LRF in our work for its simplicity and effectiveness. During the behavior control stage, at each step, we first predict a local reference frame (LRF) using subequivariant graph neural networks. We then perform canonicalization by projecting the vector states from the global world frame into the LRF as follows:

$$\mathcal{G}_{\vec{g}} \leftarrow \mathrm{LRF}(\vec{\mathcal{G}}_{\vec{g}}). \tag{4}$$

To obtain LRF, We use subequivariant graph neural networks $\varphi$ to process $\vec{\mathcal{G}}_{\vec{g}}$, predict two candidate vectors and perform orthonormalization OP:

$$\vec{\boldsymbol{u}}, \vec{\boldsymbol{v}} \leftarrow \varphi\left(\vec{\mathcal{G}}_{\vec{g}}\right), \tag{5}$$

$$\boldsymbol{O} \leftarrow \mathrm{OP}(\vec{\boldsymbol{u}}, \vec{\boldsymbol{v}}). \tag{6}$$

Specifically, $\varphi$ consists of $L$ subequivariant message passing layers. The message passing mechanism operates as follows: we first construct edge features from the node features, and then use both $\phi_{\vec{g}}$ and $\psi_{\vec{g}}$, as described in Equation (2), to generate messages. These messages are subsequently aggregated and merged with the original state. For details on subequivariant message passing, please refer to Appendix C.

After message passing, we predict $\vec{u}$ and $\vec{v}$ by using the output vector state $\vec{z}_1^{(L)}$ of the root node:

$$\vec{u} \leftarrow \vec{z}_1^{(L)} \boldsymbol{W}_{\vec{u}}, \tag{7}$$

$$\vec{v} \leftarrow \vec{z}_1^{(L)} \boldsymbol{W}_{\vec{v}}, \tag{8}$$

where $\boldsymbol{W}_{\vec{u}} \in \mathbb{R}^{m \times 1}$ and $\boldsymbol{W}_{\vec{v}} \in \mathbb{R}^{m \times 1}$.

Now we use $\vec{u}$ and $\vec{v}$ to construct a orthonormal basis. As $\vec{z}$ only consider $O_{\vec{g}}(3)$-equivariant transformations, we could set one of the axes as parallel to $\vec{g}$. We have $\boldsymbol{O} := [\vec{e}_1, \vec{e}_2, \vec{e}_3] \leftarrow \mathrm{OP}(\vec{u}, \vec{v})$:

$$\vec{e}_3 \leftarrow [0, 0, 1]^\top \tag{9}$$

$$\vec{e}_1 \leftarrow \frac{\vec{u} - \langle \vec{u}, \vec{e}_3 \rangle \vec{e}_3}{\|\vec{u} - \langle \vec{u}, \vec{e}_3 \rangle \vec{e}_3\|}, \tag{10}$$

$$\vec{e}_2 \leftarrow \frac{\vec{v} - \langle \vec{v}, \vec{e}_1 \rangle \vec{e}_1 - \langle \vec{v}, \vec{e}_3 \rangle \vec{e}_3}{\|\vec{v} - \langle \vec{v}, \vec{e}_1 \rangle \vec{e}_1 - \langle \vec{v}, \vec{e}_3 \rangle \vec{e}_3\|}. \tag{11}$$

Having obtained $\boldsymbol{O}$, we project the agent's vector state into the local reference frame to obtain the invariant vector state:

$$\boldsymbol{z}_i \leftarrow \boldsymbol{O}^\top \vec{z}_i^{(0)}, \tag{12}$$

where $\vec{z}_i^{(0)} \in \mathbb{R}^{3 \times m}$ is the input vector state of node i in the global world frame, and $z_i \in \mathbb{R}^{3 \times m}$ is the invariant vector state in the local reference frame.

Finally, we transform the subequivariant graph of the agent's state into an invariant graph $\mathcal{G}_{\vec{g}} := (\mathcal{G}_{\mathrm{m}}, \boldsymbol{H}^{\mathrm{b}}, \boldsymbol{Z}, \vec{g}) \leftarrow \mathrm{LRF}(\vec{\mathcal{G}}_{\vec{g}})$. A formal proof of the invariance of LRF Canonicalization is presented in Theorem 1.

**Behavior Control.** We use the invariant graph $\mathcal{G}_{\vec{g}}$ (such as GraphConv (Kipf & Welling, 2017)) to perform behavior control. We use a conventional neural network $\varphi_\theta^{\mathrm{b}}$ (such as MLP) to instantiate the behavior control sub-policy $\pi_\theta^{\mathrm{b}}$ and predict the behavior control action $\boldsymbol{a}^{\mathrm{b}}$:

$$\boldsymbol{a}^{\mathrm{b}} \leftarrow \varphi_\theta^{\mathrm{b}} \left( \mathcal{G}_{\vec{g}} \right). \tag{13}$$

**Value Estimation.** In the value estimation process, for greater flexibility, we employ separate individual models with the same architecture as $\varphi$ in Equation (5) to predict $\mathcal{G}_{\vec{g}}^V$. The invariant graph $\mathcal{G}_{\vec{g}}^V$ is then used for value estimation. We instantiate the value function $V_\phi$ using a conventional neural network $\varphi_\phi$ (such as MLP), which predicts the value $V$ as follows:

$$V \leftarrow \varphi_\phi \left( \mathcal{G}_{\vec{g}}^V \right). \tag{14}$$

Formal proof of the invariance of the co-evolution are presented in Appendix A.

# 4 EXPERIMENTS

## 4.1 3DS-MB ENVIRONMENTS

To evaluate our EquiEvo method, we extend the morphology-behavior co-evolution environments (Yuan et al., 2022; Huang et al., 2024), based on the MuJoCo simulation engine (Todorov et al., 2012), to operate within a practical three-dimensional physical environment. We refer to this new environment as 3DS-MB. In the 3DS-MB environments, the agent's morphology design and state are represented by the morphology graph $\mathcal{G}^{\mathrm{m}}$ and the subequivariant graph $\vec{\mathcal{G}}_{\vec{g}}$. We apply the 3DS-MB environments to three different tasks, as shown in Figure 3.

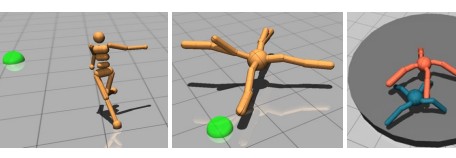

(a) (Evo)Humanoid Navigation.

(b) (Evo)Ant Navigation.

(c) (Evo)Ants Sumo.

Figure 3: Illustrations of Tasks. Navigation: the agent navigates towards the target (green). Sumo: Two agents compete against each other to push the opponent out of the arena.

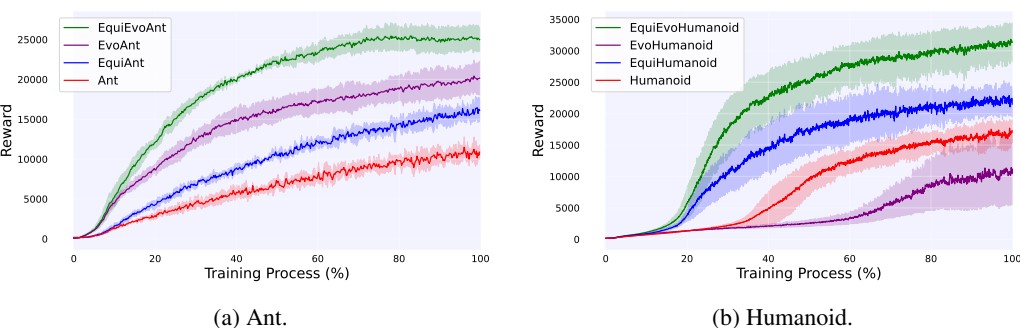

(a) Ant.                                      (b) Humanoid.

Figure 4: Training and Evaluation Curves in Navigation.

**Humanoid Navigation.  1. Initial Conditions.** The agent's position and orientation are initialized randomly, with the goal point placed randomly within a radius of $[3, 4]$ from the agent. Each time the agent reaches the goal, a new goal point is generated randomly, requiring the agent to continually adjust its direction of movement to reach as many goal points as possible. For this task, we use a humanoid morphology. To maintain the humanoid structure, we skip the structural transform stage and start directly with the attribute transform. **2. Termination.** The state must simultaneously satisfy isfinite($\vec{\boldsymbol{p}}$), isfinite($\vec{\boldsymbol{v}}$), and $h \in [\text{minHeight}, \text{maxHeight}]$, where $\vec{\boldsymbol{p}}$, $\vec{\boldsymbol{v}}$, and $h$ represent the position, velocity, and height of the agent, respectively. The values minHeight and maxHeight are the preset minimum and maximum heights. If any of these conditions are not met, the training is terminated. For the humanoid agent, we set $h \in [1, 2]$. **3. Reward.** For the navigation task, we design a reward structure for the humanoid agent that consists of six components: *success bonus, forward reward, distance reward, control cost, contact cost,* and *survive reward.* The total reward is the sum of these components. For more details, please refer to Appendix E.1.

**Ant Navigation.  1. Initial Conditions.** The initial setup is similar to that of the Humanoid Navigation task. In this task environment, we use the ant morphology for experimentation, incorporating the structure transform stage, attribute transform stage, and behavior control stage during the evolution process. **2. Termination.** The termination conditions mirror those of the Humanoid Navigation task. For the ant agent, we set $h \in [0.28, 0.8]$. **3. Reward.** For the navigation task, we design a reward structure for the Ant agent consisting of four components: *success bonus, distance reward, control cost,* and *survive reward.* The total reward is the sum of these components. For further details, please refer to Appendix E.1.

**Ants Sumo.  1. Initial Conditions.** The positions and orientations of two agents are randomly initialized, requiring them to learn to attack and defend in various strategic situations while attempting to push each other out of the arena. The radius of the arena is randomized within the range of $[2.5, 4.5]$, and the arena has a fixed height of 0.5. Additionally, the agents are permitted to reach a maximum height of 0.29 above the arena ground. In this environment, we use the ant morphology for experimentation. Following the CompetEvo (Huang et al., 2024) setup, the structure transform stage is omitted, retaining only the attribute transform stage and behavior control stage. **2. Termination.** The objective is for one agent to win by disqualifying the opponent, which occurs when an agent pushes the other out of the arena or when an agent's height exceeds the set limit (draw outcomes). The disqualified agent loses, ending the episode. **3. Reward.** For the Ants Sumo task, we design a reward structure consisting of seven components: *win bonus, lose bonus, draw bonus, control cost, push opponent reward, move reward,* and *survive reward.* The total reward is the sum of these components. For more details, please refer to Appendix E.1.

We chose to evolve morphology skeletons from scratch to address the most challenging setup and eliminate biases from prior knowledge, starting the Ant Navigation task with a simple "atomic morphology" (a torso body). For other starting morphologies, such as in the Humanoid Navigation and Ants Sumo tasks, we focused solely on attribute transformations to maintain task-specific constraints, as structural transformation boundaries are less well-defined.

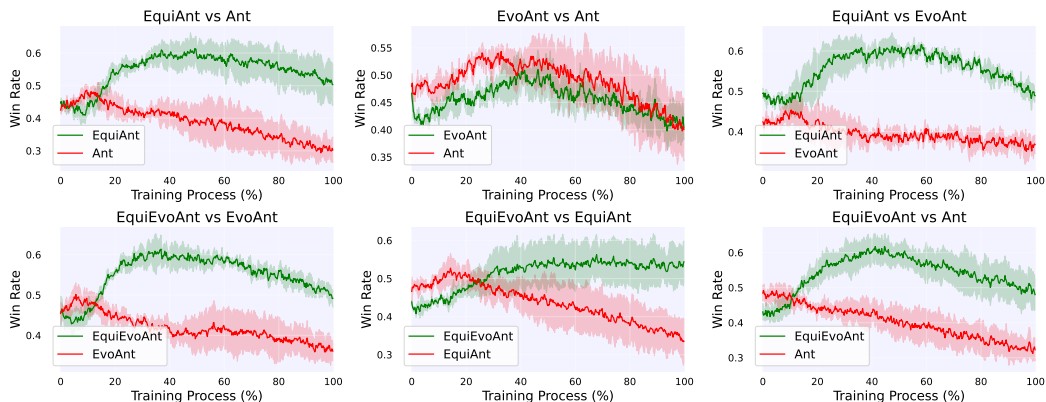

Figure 5: Training and Evaluation Curves for Sumo.

## 4.2 BASELINES, METRICS, AND IMPLEMENTATIONS

**Baselines.** For Agent X(Ant/Humanoid), we employ the EquiEvo method to obtain EquiEvoX. To conduct a comprehensive comparison, we define the other three baseline methods: EquiX, EvoX, X. EquiX maintains subequivariance but removes the morphology transform component. EvoX includes morphology transform but does not integrate subequivariance. X is the most basic baseline method, lacking both subequivariance and morphology transform component. For the three types of tasks, the specific methods adopted are as follows:

- Ant Navigation: EquiEvoAnt, EquiAnt, EvoAnt, Ant.
- Humanoid Navigation: EquiEvoHumanoid, EquiHumanoid, EvoHumanoid, Humanoid.
- Ants Sumo: EquiEvoAnt, EquiAnt, EvoAnt, Ant.

The Ant Navigation uses the Transform2Act (Yuan et al., 2022) codebase as the baseline, while the Humanoid Navigation and Ant Sumo use the CompetEvo (Huang et al., 2024) codebase as the baseline. For more information on the baselines, please refer to Appendix E.2.

**Metrics.** For Navigation task, we use cumulative reward to assess the performance of the agents; for Sumo task, we use win rate (*Win Rate* = #*win_episode* / #*all_episode*). Each experiment is trained with 3 seeds to report the average and standard deviation of the cumulative reward or win rate.

**Implementations.** We use PPO (Schulman et al., 2017) as the reinforcement learning algorithm in all experiments. We implemented EquiEvo based on the Transform2Act codebase (Yuan et al., 2022) and the CompetEvo codebase (Huang et al., 2024), which are built on the PyTorch framework. There is no weight sharing between the policy network $\pi_\theta$ and the value estimation network $V_\phi$. The value of the maximum timesteps of an episode is 1,000 (Navigation) / 500 (Sumo). The value of the maximum epoch is 500 (Ant) / 2,000 (Humanoid). Additionally, in Sumo, we adopt Bansal et al. (2018)'s approach, where teams using different methods compete against each other within an arena. For detailed hyperparameters, refer to Table 2.

## 4.3 MAIN RESULTS

**Navigation.** We first conduct a performance comparison of methods on the navigation task, as shown in Figure 4.

For the Humanoid agent, *EquiEvo* surpasses baseline methods across random scenarios (Figure 4b). The ranking from best to worst—*EquiEvoHumanoid*, *EquiHumanoid*, *Humanoid*, and *EvoHumanoid*—highlights the significance of subequivariance in morphology-behavior co-evolution. Without subequivariance, applying morphology transform alone (*Evo*) expands the search space,



Figure 6: Visualization of Humanoid Morphology.

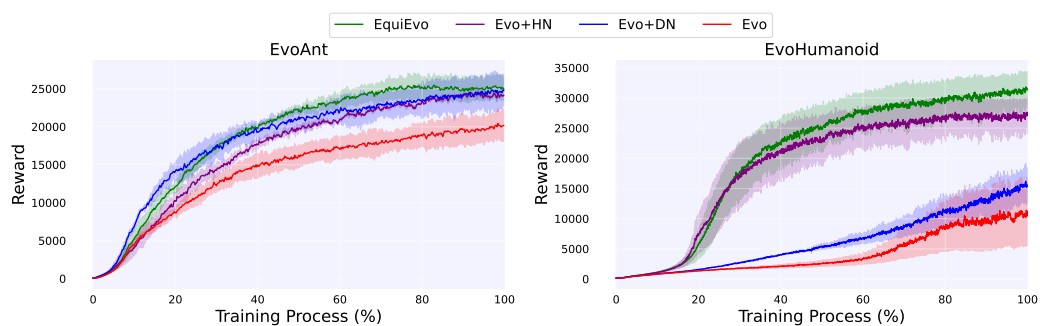

Figure 7: Comparisons with Hand-Craft Normalization.

hindering efficient training. This demonstrates the crucial role of subequivariance in optimizing co-evolution.

The visualization in Figure 6 demonstrates that subequivariance significantly impacts the shape of the humanoid agent. *EvoHumanoid*, developed without geometric symmetry constraints, lacks balance, resulting in poor coordination and stability during movement. In contrast, *EquiEvoHumanoid* develops a more coordinated morphology, leading to greater stability and flexibility—more akin to that of a professional athlete.

For Ant, the experimental results (Figure 4a) rank *EquiEvoAnt* as the best in locomotion navigation, followed by *EvoAnt*, *EquiAnt*, and *Ant*. The *EquiEvo* method enhances learning by integrating subequivariance and morphology transform, demonstrating the effectiveness of both components in our approach. *EvoAnt* performs well due to its morphology transform, while *EquiAnt* is weaker without it but still outperforms *Ant*.

**Sumo.** In the sumo task (Figure 5), the *EquiEvoAnt* agent consistently outperformed all baselines. The results emphasize the critical impact of subequivariance, which proved more influential than morphology evolution alone in enhancing performance. However, integrating morphology evolution with subequivariance yielded a synergistic effect, further boosting the agents' capabilities. This highlights that subequivariance is key to success in sumo tasks, with the addition of morphology evolution refining and optimizing strategies for significant performance gains.

## 4.4 Ablations and Analyses

**Comparisons with Hand-Crafted Normalization.** In addition to our equivariant network approach, hand-crafted methods can also construct a Local Reference Frame (LRF). To compare the effectiveness of these approaches, we designed the following experiments, ablated in Figure 7: **1. Evo+DN**: a variant using hand-crafted normalization where $\vec{z}$ are treated as scalars and the goal direction is used to construct the LRF; **2. Evo+HN**: another hand-crafted normalization variant, treating $\vec{z}$ as scalars but constructing the LRF based on the agent's heading direction; **3. Evo**: a non-equivariant variant that treats $\vec{z}$ as scalars without any LRF construction.

*Can equivariant network methods replace or even surpass hand-crafted normalization?* Experiments demonstrate that regardless of using goal or heading direction to construct the LRF, hand-crafted methods perform worse than ours, indicating that equivariant networks are more effective for morphology evolution, offering a plug-in solution for equivariant adaptation.

*Do different tasks require different LRF designs for symmetry?* We observe that EvoAnt, a goal-directed task, benefits more from DN since the primary reward comes from approaching the goal. In contrast, the humanoid task, which requires both goal-reaching and forward motion, sees better performance with HN. This suggests that optimal LRF design varies by task. Additionally, evidences from prior works (Chen et al., 2023a; 2024) indicate that learning-based equivariant methods, which automatically identify the optimal LRF by leveraging geometric information, outperform hand-crafted approaches that rely on manually applying symmetry priors.

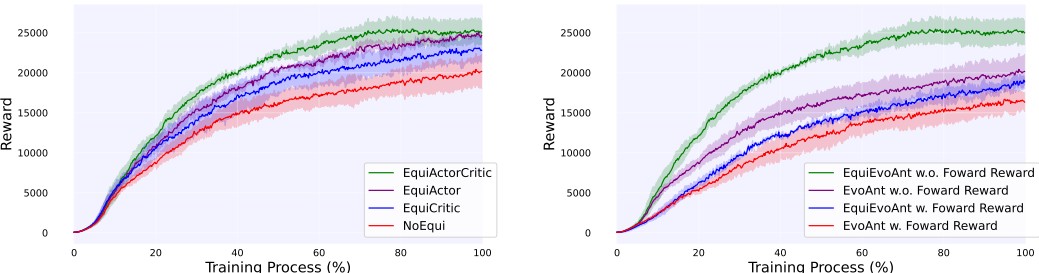

Figure 8: Ablations of Equivariance on EvoAnt.  Figure 9: Analyses of Morphology-Task.

**Ablations of Equivariance.** In *EquiEvo* (or *EquiActorCritic*), both the Actor and Critic networks are designed to be subequivariant, ensuring that the entire reinforcement learning process adheres to equivariance under the Actor-Critic framework. To evaluate the impact of each network, we perform ablation experiments where one network retains subequivariance while the other does not, producing the variants *EquiActor* and *EquiCritic*, respectively. The results in Figure 8 demonstrate that the performance of *EquiEvo* diminishes when either the Actor or Critic lacks subequivariance, emphasizing the necessity of this property in both networks. Moreover, the Actor's subequivariance appears to play a more crucial role than the Critic's, likely because of its direct impact on decision-making and action selection. These findings underscore the importance of maintaining subequivariance in both the Actor and Critic to achieve an efficient and consistent reinforcement learning process.

**Analyses of Morphology-Task Mapping.** The evolved morphology of *EquiEvoAnt* exhibits strong symmetry, as seen in Figure 10, which coincides with the symmetric structure of the task reward. To investigate how task design affects morphology evolution, we introduce a forward reward to *EvoAnt* and train both *EquiEvoAnt*

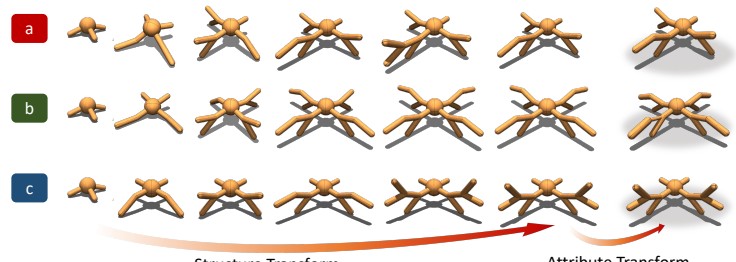

Figure 10: Step-by-step visualization of the morphology transformation within a single episode, corresponding to the final checkpoint . (a) EvoAnt; (b) EquiEvoAnt; (c) EquiEvoAnt with Forward Reward.

and *EvoAnt* under this modified reward structure. The reward curves, presented in Figure 9, indicate that *EquiEvoAnt* continues to outperform *EvoAnt* in this altered task. In Figure 10 (b) and (c), we compare the evolution processes under different task (reward) settings. For the task with a forward reward, the evolved morphology exhibits a laterally symmetric structure with stronger front legs and weaker hind legs, while the task without a forward reward leads to an radially symmetric morphology. This shows that the evolution of morphology is fundamentally shaped by environmental interactions and task demands, rather than being a predefined goal. The comparison between Figure 10 (a) and (b) shows that methods without integrating geometric equivariance, due to lower sample efficiency, fail to fully evolve a radially symmetric morphology that meets task requirements. This highlights the necessity of considering geometric equivariance in the co-evolution of morphology and behavior frameworks.

**Analysis of Morphology-Behavior Mapping.** We conduct experiments on the Ant Navigation task to analyze robust Morphology-Behavior Mapping, with details in Appendix E.3.

## 5 CONCLUSIONS

We present EquiEvo, a 3D subequivariant framework for co-evolving morphology and behavior, leveraging geometric symmetries to enhance efficiency and adaptability. Our results demonstrate its effectiveness in navigation and sumo tasks, highlighting the importance of geometric equivariance in driving task-specific morphologies through interaction-driven optimization. Further discussion on structural evolution is provided in Appendix E.4. Future work can explore more complex environments, tasks, and evolution settings to further advance embodied intelligence and robotic design.

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

## A  PROOFS

In this section, we theoretically prove that our proposed EquiEvo ensures the co-evolution process preserve the symmetry as desired.

Finally, we transform the subequivariant graph of the agent's state into an invariant graph

**Theorem 1.** *The LRF canonicalization, denoted as $\mathcal{G}_{\vec{g}} \leftarrow \mathrm{LRF}(\vec{\mathcal{G}}_{\vec{g}})$ is $\mathrm{O}_g(3)$-invariant, satisfying any transformation $g \in \mathrm{O}_{\vec{g}}(3)$, $\mathcal{G}_{\vec{g}} = \mathrm{LRF}(g \cdot \vec{\mathcal{G}}_{\vec{g}})$.*

*Proof.* Let the projected graph $\mathcal{G}_{\vec{g}}$ be the output of the LRF canonicalization with input $\vec{\mathcal{G}}_{\vec{g}} = (\mathcal{G}_{\mathrm{m}}, \boldsymbol{H}^{\mathrm{b}}, \vec{\boldsymbol{Z}}, \vec{g})$. Similarly, let $\mathcal{G}'_{\vec{g}}$ be the output of the LRF canonicalization with input $g \cdot \vec{\mathcal{G}}_{\vec{g}} = (\mathcal{G}_{\mathrm{m}}, \boldsymbol{H}^{\mathrm{b}}, g \cdot \vec{\boldsymbol{Z}}, \vec{g})$, for any transformation $g \in \mathrm{O}_{\vec{g}}(3)$. If $\mathcal{G}_{\vec{g}} = \mathcal{G}'_{\vec{g}}$, this indicates that the LRF canonicalization preserves $\mathrm{O}_{\vec{g}}(3)$-invariance.

We first prove that OP in Equation (6) is equivariant.

Let $g$ be a transformation in $\mathrm{O}_{\vec{g}}(3)$, which includes rotation and reflection $R$ along the direction of $\vec{g}$. Specifically, the transformation is applied as follows:

$$\boldsymbol{O}' \leftarrow g \cdot \boldsymbol{O} := R\boldsymbol{O}_i, \tag{15}$$

$$\boldsymbol{z}' \leftarrow g \cdot \vec{\boldsymbol{v}} := R\vec{\boldsymbol{z}}. \tag{16}$$

We use the fact that $\forall R \in \mathrm{O}_{\vec{g}}(3), \forall \vec{\boldsymbol{x}}, \vec{\boldsymbol{y}} \in \mathbb{R}^3, \langle R\vec{\boldsymbol{x}}, R\vec{\boldsymbol{y}} \rangle = \vec{\boldsymbol{x}}^\top R^\top R \vec{\boldsymbol{y}} = \vec{\boldsymbol{x}}^\top \vec{\boldsymbol{y}} = \langle \vec{\boldsymbol{x}}, \vec{\boldsymbol{y}} \rangle$ and $\|R\vec{\boldsymbol{x}}\| = \langle R\vec{\boldsymbol{x}}, R\vec{\boldsymbol{x}} \rangle = \langle \vec{\boldsymbol{x}}, \vec{\boldsymbol{x}} \rangle = \|\vec{\boldsymbol{x}}\|$. By definition, $R\vec{e}_3 = \vec{e}_3$. According to the property of Equation (5), we know that $\vec{\boldsymbol{z}}_1^{(L)}$, $\vec{\boldsymbol{u}}$ and $\vec{\boldsymbol{v}}$ are $\mathrm{O}_{\vec{g}}(3)$-equivariant.

Let $\boldsymbol{O}' \leftarrow \mathrm{OP}(g \cdot \vec{\boldsymbol{u}}, g \cdot \vec{\boldsymbol{v}})$. By the properties of the orthogonalization process in Equations (9) to (11), we have:

$$\vec{e}_3' = [0, 0, 1]^\top = \vec{e}_3 = R\vec{e}_3, \tag{17}$$

$$\vec{e}_1' = \frac{R\vec{\boldsymbol{u}} - \langle R\vec{\boldsymbol{u}}, \vec{e}_3' \rangle \vec{e}_3'}{\|R\vec{\boldsymbol{u}} - \langle R\vec{\boldsymbol{u}}, \vec{e}_3' \rangle \vec{e}_3'\|} \tag{18}$$

$$= \frac{R\vec{\boldsymbol{u}} - \langle R\vec{\boldsymbol{u}}, R\vec{e}_3 \rangle R\vec{e}_3}{\|R\vec{\boldsymbol{u}} - \langle R\vec{\boldsymbol{u}}, R\vec{e}_3 \rangle R\vec{e}_3\|} \tag{19}$$

$$= \frac{R(\vec{\boldsymbol{u}} - \langle \vec{\boldsymbol{u}}, \vec{e}_3 \rangle \vec{e}_3)}{\|R(\vec{\boldsymbol{u}} - \langle \vec{\boldsymbol{u}}, \vec{e}_3 \rangle \vec{e}_3)\|} \tag{20}$$

$$= R\frac{(\vec{\boldsymbol{u}} - \langle \vec{\boldsymbol{u}}, \vec{e}_3 \rangle \vec{e}_3)}{\|(\vec{\boldsymbol{u}} - \langle \vec{\boldsymbol{u}}, \vec{e}_3 \rangle \vec{e}_3)\|} \tag{21}$$

$$= R\vec{e}_1, \tag{22}$$

$$\vec{e}_2' = \frac{R\vec{\boldsymbol{v}} - \langle R\vec{\boldsymbol{v}}, \vec{e}_1' \rangle \vec{e}_1' - \langle R\vec{\boldsymbol{v}}, \vec{e}_3' \rangle \vec{e}_3'}{\|R\vec{\boldsymbol{v}} - \langle R\vec{\boldsymbol{v}}, \vec{e}_1' \rangle \vec{e}_1' - \langle R\vec{\boldsymbol{v}}, \vec{e}_3' \rangle \vec{e}_3'\|} \tag{23}$$

$$= \frac{R\vec{\boldsymbol{v}} - \langle R\vec{\boldsymbol{v}}, R\vec{e}_1 \rangle R\vec{e}_1 - \langle R\vec{\boldsymbol{v}}, R\vec{e}_3 \rangle R\vec{e}_3}{\|R\vec{\boldsymbol{v}} - \langle R\vec{\boldsymbol{v}}, R\vec{e}_1 \rangle R\vec{e}_1 - \langle R\vec{\boldsymbol{v}}, R\vec{e}_3 \rangle R\vec{e}_3\|} \tag{24}$$

$$= \frac{R(\vec{\boldsymbol{v}} - \langle \vec{\boldsymbol{v}}, \vec{e}_1 \rangle \vec{e}_1 - \langle \vec{\boldsymbol{v}}, \vec{e}_3 \rangle \vec{e}_3)}{\|R(\vec{\boldsymbol{v}} - \langle \vec{\boldsymbol{v}}, \vec{e}_1 \rangle \vec{e}_1 - \langle \vec{\boldsymbol{v}}, \vec{e}_3 \rangle \vec{e}_3)\|} \tag{25}$$

$$= R\frac{(\vec{\boldsymbol{v}} - \langle \vec{\boldsymbol{v}}, \vec{e}_1 \rangle \vec{e}_1 - \langle \vec{\boldsymbol{v}}, \vec{e}_3 \rangle \vec{e}_3)}{\|(\vec{\boldsymbol{v}} - \langle \vec{\boldsymbol{v}}, \vec{e}_1 \rangle \vec{e}_1 - \langle \vec{\boldsymbol{v}}, \vec{e}_3 \rangle \vec{e}_3)\|} \tag{26}$$

$$= R\vec{e}_2. \tag{27}$$

Therefore,

$$\boldsymbol{O}' = [\vec{e}_1', \vec{e}_2', \vec{e}_3'] = R[\vec{e}_1, \vec{e}_2, \vec{e}_3] = R\boldsymbol{O}. \tag{28}$$

Then we have

$$\boldsymbol{z}_i' = \boldsymbol{O}'^\top \vec{\boldsymbol{z}}_i^{(0)\prime} = \boldsymbol{O}^\top R^\top R\vec{\boldsymbol{z}}_i^{(0)} = \boldsymbol{O}^\top \vec{\boldsymbol{z}}_i^{(0)} = \boldsymbol{z}_i. \tag{29}$$

Hence,

$$\mathcal{G}'_{\vec{g}} = (\mathcal{G}_{\mathrm{m}}, \boldsymbol{H}^{\mathrm{b}}, \boldsymbol{Z}', \vec{g}) \tag{30}$$

$$= (\mathcal{G}_{\mathrm{m}}, \boldsymbol{H}^{\mathrm{b}}, \boldsymbol{Z}, \vec{g}) \tag{31}$$

$$= \mathcal{G}_{\vec{g}}. \tag{32}$$

$\square$

**Corollary 1.** *Let $\boldsymbol{a}_t^{\mathrm{m}}, \boldsymbol{a}_t^{\mathrm{b}}, V_\phi$ be output of the actor and the critic of EquiEvo with $\mathcal{G}^{\mathrm{m}}$ and $\vec{\mathcal{G}}_{\vec{g}}$ as input. Let $\boldsymbol{a}_t^{\mathrm{m}\prime}, \boldsymbol{a}_t^{\mathrm{b}\prime}, V_\phi'$ be the actor and critic with $g \cdot \mathcal{G}^{\mathrm{m}}$ and $g \cdot \vec{\mathcal{G}}_{\vec{g}}$ as input, $g \in O_{\vec{g}}(3)$. Then, $(\boldsymbol{a}_t^{\mathrm{m}\prime}, \boldsymbol{a}_t^{\mathrm{b}\prime}, V_\phi') = (\boldsymbol{a}_t^{\mathrm{m}}, \boldsymbol{a}_t^{\mathrm{b}}, V_\phi)$, indicating the EquiEvo preserve $O_{\vec{g}}(3)$-invariance.*

*Proof.* First, based on the geometric transformation properties of the topology graph, we have

$$\mathcal{G}^{\mathrm{m}\prime} = g \cdot \mathcal{G}^{\mathrm{m}} = \mathcal{G}^{\mathrm{m}}. \tag{33}$$

Then, by Theorem 1, $\forall R \in \boldsymbol{O}_{\vec{g}}(3)$, we have

$$\mathcal{G}'_{\vec{g}} = g \cdot \vec{\mathcal{G}}_{\vec{g}} = \mathcal{G}_{\vec{g}}. \tag{34}$$

Therefore,

$$\boldsymbol{a}_t^{\mathrm{m}\prime} = \varphi_\theta^{\mathrm{m}}(\mathcal{G}^{\mathrm{m}\prime}) \tag{35}$$

$$= \varphi_\theta^{\mathrm{m}}(\mathcal{G}^{\mathrm{m}}) = \boldsymbol{a}_t^{\mathrm{m}}, \tag{36}$$

$$\boldsymbol{a}_t^{\mathrm{b}\prime} = \varphi_\theta^{\mathrm{b}}(\mathcal{G}'_{\vec{g}}) \tag{37}$$

$$= \varphi_\theta^{\mathrm{b}}(\mathcal{G}'_{\vec{g}}) = \boldsymbol{a}_t^{\mathrm{b}}, \tag{38}$$

and

$$V_\phi' = \varphi_\phi^{\mathrm{b}}(\mathcal{G}'_{\vec{g}}) \tag{39}$$

$$= \varphi_\phi^{\mathrm{b}}(\mathcal{G}'_{\vec{g}}) = V_\phi. \tag{40}$$

$\square$

## B  RELATED WORK

**Optimization of Agent morphology Design:** Research in the field of optimization of agent morphology design mainly falls into two mainstream approaches: the first focuses on optimizing the properties, functions, and design parameters of agents (Schaff et al., 2019; Ha, 2019; Chen et al., 2023b), and the second on structure optimization based on graph neural networks (Wang et al., 2018; Yuan et al., 2022; Hu et al., 2023). The former often adopts population-based methods, handling morphology and control strategy optimization separately through a two-level optimization process (Ha, 2019), such as the application of evolutionary search in morphology optimization (Gupta et al., 2021; Wang et al., 2019); the latter focuses on multi-objective joint optimization (Yuan et al., 2022). Research also includes the optimization of continuous design parameters of agents, using methods such as simulated annealing, trajectory optimization, and deep reinforcement learning (Baykal & Alterovitz, 2017; Ha et al., 2017; Chen et al., 2020). Recent studies have further simplified the evolutionary process and improved task-independent design optimization (Cheney et al., 2014; Desai et al., 2017; III et al., 2021). Despite the progress made by these methods, sample efficiency remains a challenge, as agents with different design morphologies often learn independently within the population. Yuan et al. (2022) introduce Transform2Act, an innovative policy for simultaneous morphology design and control optimization that adapts skeletal structures and joint attributes for efficient environmental interaction, advancing integrated agent design and control. Huang et al. (2024) presents CompetEvo, a system that integrates morphology evolution with adversarial game self-training to enhance agents' capabilities, showing morphology evolution significantly boosts combat skills and emergent behaviors.

**Geometric Equivariant Models:** Notably, the physical realm exhibits inherent symmetries, and extensive research has been conducted on group equivariant models (Cohen & Welling, 2016a;b; Worrall et al., 2017). In recent times, the research area of geometrically equivariant graph neural networks (Han et al., 2022b) has emerged, utilizing symmetry as an inductive bias during the learning process. These models are crafted to ensure that their outputs undergo rotation, translation, or reflection in alignment with the inputs, thereby preserving the symmetry. Within a Markov decision process (MDP) that exhibits symmetries (van der Pol et al., 2020), the state-action space possesses inherent symmetries, allowing policies to be optimized within a more straightforward, abstract MDP. van der Pol et al. (2020) endeavors to train policy networks that are equivariant and value networks that are invariant in two-dimensional simulated environments. In contrast, Chen et al. (2023a) delves into the study of body-level equivariant policy networks within intricate 3D physics environments, enhancing policy generalization over various orientations. Additionally, Chen et al. (2024) address the challenges of multi-agent games by employing entity assignment along with an entity-level subequivariant message-passing mechanism. Both our approach and Chen et al. (2023a; 2024) implement equivariance based on the core function described in Equation (2). However, there are key differences in the design and application. Chen et al. (2023a) employs a transformer architecture with a fully connected topology, whereas both Chen et al. (2024) and our approach utilize GNNs. The main distinction lies in the graph construction: Chen et al. (2024) models multi-agent relationships, requiring an entity assignment to define the graph's topology, while our method focuses on single-agent morphology evolution, directly using the morphology's inherent topology to build the graph. This design aligns with our goal of co-optimizing morphology and behavior efficiently within a single-agent framework.

## C SUBEQUIVARIANT MESSAGE PASSING

Concretely, we have:

$$\vec{z}_{ij}^{(l)} = [(\vec{p}_j - \vec{p}_i), \vec{z}_i^{(l)}, \vec{z}_j^{(l)}], \tag{41}$$

$$h_{ij}^{(l)} = [\|\vec{p}_j - \vec{p}_i\|_2, h_i^{(l)}, h_j^{(l)}], \tag{42}$$

$$\vec{m}_{ij}^{(l)}, m_{ij}^{(l)} = \phi_{\vec{g}}\left(\vec{z}_{ij}^{(l)}, h_{ij}^{(l)}\right), \tag{43}$$

$$\vec{m}_i^{(l)}, m_i^{(l)} = \sum_{j \in \mathcal{N}(i)} \vec{m}_{ij}^{(l)}, \sum_{j \in \mathcal{N}(i)} m_{ij}^{(l)}, \tag{44}$$

$$(\vec{z}_i^{(l+1)}, h_i^{(l+1)}) = (\vec{z}_i^{(l)}, h_i^{(l)}) + \psi_{\vec{g}}\left([\vec{m}_i^{(l)}, \vec{z}_i^{(l)}], [m_i^{(l)}, h_i^{(l)}]\right), \tag{45}$$

where

- $\vec{p}_i$, the position of node $i$;
- $h_i^{(l)}$ and $\vec{z}_i^{(l)}$, the scalar and vector states of node $i$ at the $l$-th layer;
- $h_{ij}^{(l)}$ and $\vec{z}_{ij}^{(l)}$, the scalar and vector states of edge $ij$ at the $l$-th layer;
- $\vec{m}_{ij}^{(l)}$ and $m_{ij}^{(l)}$, the vector and scalar messages passed through edge $ij$;
- $\vec{m}_i^{(l)}$ and $m_i^{(l)}$, the aggregated vector and scalar messages at node $i$;
- $\phi_{\vec{g}}$ and $\psi_{\vec{g}}$, as described in Equation (2), the subequivariant functions used for message passing and state updates.

## D THE ROLE OF GEOMETRIC SYMMETRY IN MORPHOLOGY EVOLUTION

The optimization of morphology in co-evolution tasks is governed by a bi-level objective, as defined in Equation (1):

$$\mathcal{G}_{\mathrm{m}}^* := \arg\max_{\mathcal{G}_m} J(\pi_{\mathcal{G}}, \mathcal{G}_{\mathrm{m}}) \quad \text{s.t.} \quad \pi_{\mathcal{G}} = \arg\max_{\pi} J(\pi, \mathcal{G}_{\mathrm{m}}),$$

where $J(\pi, \mathcal{G}_{\mathrm{m}})$ represents the cumulative reward for a given morphology $\mathcal{G}_{\mathrm{m}}$ and policy $\pi$. This two-layer optimization highlights the dependence of morphology optimization on the behavior policy that performs for a given morphology. It is important to note that $J$ is defined as an expectation,

meaning that all states are integrated out in the expectation sense. Consequently, the morphology's policy and value functions theoretically depend only on the morphology $\mathcal{G}_{\mathrm{m}}$ and the behavior policy $\pi_{\mathcal{G}}$.

In practice, the bi-level optimization is implemented as a co-optimization process, where the inner and outer optimizations alternate incrementally rather than fully optimizing inner layer. Since the evaluation of $J$ relies on finite samples of the behavior policy, if the behavior policy is not equivariant, the limited sampling will cause $J$ to vary under sampled initial state. Geometric symmetry reduces the variability introduced by sampling, as equivariant networks ensure equivalent actions and consistent values, providing consistent feedback across state transformations——such as translations, rotations, or reflections. For instance, under rotational symmetry, equivalent actions are treated as identical, and the feedback provided to the morphology is unaffected by differences in sampled initial states. This consistent feedback not only improves the reliability of morphology evaluation but also enhances the efficiency of the co-evolution process, allowing for better optimization within the search space.

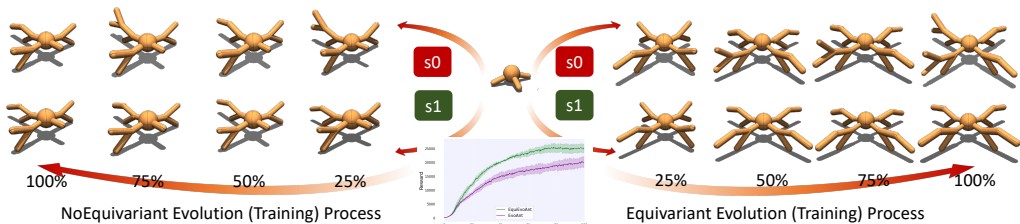

Figure 11: Visualization of Morphological Evolution Across Training Progress.

Figure 11 compares the morphological evolution during training under two scenarios: without (left) and with (right) the integration of geometric symmetry. Checkpoints at 25%, 50%, 75%, and 100% of the training process are visualized for two random seeds (s0 and s1). Without injecting geometric symmetry (left), morphology evolution appears asymmetric and incomplete, with structural transformations seemingly getting trapped in local optima early in the process, leading to lower reward curves. This may be attributed to the nature of reinforcement learning, where morphology and behavior are gradually evolved through abundant exploration of the environment. During this learning process, the co-evolution of morphology and behavior becomes highly vulnerable to local minima, and searching for a good policy within the large space would be notoriously difficult. **In contrast, injecting geometric symmetry (right) fosters more intricate and comprehensive morphology evolution, accompanied by higher reward curves, indirectly supporting the claim that geometric symmetry enhances the efficiency of the co-evolution process by compacting the search space redundancy in a lossless manner, enabling better optimization.**

## E    MORE EXPERIMENTALS DETAILS AND RESULTS

### E.1    REWARD

**Humanoid Navigation:** For the navigation task, we design the reward for Humanoid agent. The designed reward structure comprises six components: **a. Success Bonus:** A significant sparse reward of 1,000 is awarded. **b. Forward Reward:** It is quantified as $1.25 \times \frac{\frac{\vec{p}}{||\vec{p}||} \cdot \Delta \vec{p}}{\Delta t}$, where $\frac{\vec{p}}{||\vec{p}||}$ represents the unit direction vector of the agent's position, and $\Delta \vec{p}$ represents the displacement of the agent from the current time step. **c. Distance Reward:** A dense reward to incentivize the achievement of the task objective. It is computed as $\frac{\Delta s}{\Delta t}$, where $\Delta s$ represents the change in distance from the agent to the target. **d. Control Cost:** This penalty discourages agents from executing excessively large actions, and is calculated as $-0.1 \times \sum \sqrt{a_i}$. **e. Contact Cost:** The penalty discourages excessive contact forces, and is calculated as $-\min(0.5 \times 10^{-6} \times \sum c_i^2, 10)$, where $c_i$ represents the component of the force exerted on the agent by external forces. **f. Survive Reward:** For each step taken, the agent receives a survival reward of 5. Thus, the reward is equal to the sum of the aforementioned rewards.

**Ant Navigation:** For the navigation task, we design the reward for Ant agent. The designed reward structure comprises four components: **a. Success Bonus:** A significant sparse reward of 1,000 is awarded. **b. Distance Reward:** A dense reward to incentivize the achievement of the task objective. It is computed as $10 \times \frac{\Delta s}{\Delta t}$, where $\Delta s$ represents the change in distance from the agent to the target. **c. Control Cost:** This penalty discourages agents from executing excessively large actions, and is calculated as $-0.01 \times \sum \sqrt{a_i}$. **d. Survive Reward:** For each step taken, the agent receives a survival reward of 1. Thus, the reward is equal to the sum of the aforementioned rewards.

**Ants Sumo:** For Ants Sumo task, we design the reward structure. The designed reward structure comprises seven components: **a. Win Bonus:** A sparse reward of 2000 for achieving the win condition. **b. Lose Bonus:** A sparse penalty of -2000 for the losing condition. **c. Draw Bonus:** A sparse penalty of -1000 for the draw condition. This penalty is meant to encourage confrontation. **d. Control Cost:** This penalty discourages agents from executing excessively large actions, and is calculated as $-0.1 \times \sum \sqrt{a_i}$. **e. Push Opponent Reward:** This motivates agents to push the opponent, calculated as $-10 \times \exp(-||\vec{p}_{opp}||)$, where $\vec{p}_{opp}$ represents the position of the opponent. **f. Move Reward:** This motivates agents to move closer to the opponent, calculated as $10 \times \max(\Delta\vec{p} \cdot \frac{\vec{p}_{opp}-\vec{p}}{||\vec{p}_{opp}-\vec{p}||}, 0)$, where $\Delta\vec{p}$ represents the displacement of the agent from the current time step. **g. Survive Reward:** For each step taken, the agent receives a survival reward of 2. Thus, the reward is equal to the sum of the aforementioned rewards.

## E.2 BASELINE METHODS

**Ant Navigation:** In the navigation task, we use Ant agent for experimentation. The agent we improve by using EquiEvo method is named EquiEvoAnt, which integrates both subequivariance and morphology evolution mechanism. To conduct a comprehensive comparison, we define the following three baseline methods: EquiAnt: This baseline method maintains subequivariance but removes the morphology evolution component. EvoAnt: This baseline method includes morphology evolution but does not integrate subequivariance. Ant: This is the most basic baseline method, lacking both subequivariance and morphology evolution component.

**Humanoid Navigation:** In the navigation task, we also introduce Humanoid agent for experimentation, while we omit structure transform stage during evolution. After applying the EquiEvo method, we name the agent EquiEvoHumanoid. To conduct a comprehensive comparison, we define the following three baseline methods: EquiHumanoid: This baseline method maintains subequivariance but removes the morphology evolution component. EvoHumanoid: This baseline method includes morphology evolution but does not integrate subequivariance. Humanoid: This is the most basic baseline method, lacking both subequivariance and morphology evolution capabilities. In this task, we skip structure transform stage.

**Ants Sumo:** In the sumo task, we still use the Ant agent for experimentation. We adopt the EquiEvo method and omit the structure transform stage during evolution, naming it EquiEvoAnt, which integrates both subequivariant networks and morphology evolution mechanisms. To conduct a comprehensive comparison, we define the following three baseline methods: EquiAnt: This baseline method maintains subequivariance but removes the morphology evolution component. EvoAnt: This baseline method includes morphology evolution but does not integrate subequivariance. Ant: This is the most basic baseline method, lacking both subequivariance and morphology evolution capabilities. In this task, we skip structure transform stage.

## E.3 ANALYSIS OF MORPHOLOGY-BEHAVIOR MAPPING

We conduct experiments on the Ant Navigation task to analyze the impact of Morphology-Behavior Mapping, particularly focusing on the role of JSMLP, which was proposed in Transform2Act (Yuan et al., 2022) to differentiate joint actions. While JSMLP introduces body-specific characteristics, enhancing effectiveness in some contexts, it reduces the generality of the morphology-behavior mapping and impedes reusability in tasks with symmetric structures. To validate this, we ablate four variants: *Evo w. JSMLP* (baseline with JSMLP), *Evo w.o. JSMLP* (baseline without JSMLP), *EquivEvo w.o. JSMLP* (our method without JSMLP), and *EquivEvo w. JSMLP* (our method with JSMLP). As shown in Figure 12, the removal of the JSMLP module improves performance, with *EquivEvo w.o. JSMLP* outperforming all variants. This indicates that eliminating JSMLP enhances

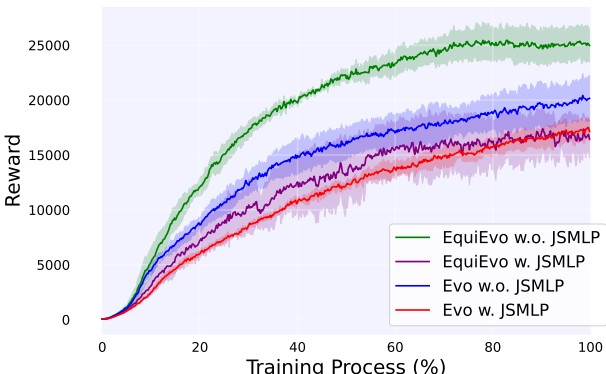

Figure 12: Analyses of Morphology-Behavior.

adaptability and learning by maintaining the generality of the morphology-behavior mapping, allowing better exploitation of task symmetry. Consequently, simplifying the model by omitting JSMLP within the *EquivEvo* framework facilitates more efficient navigation in symmetric tasks.

### E.4    DISCUSSION ON STRUCTURAL EVOLUTION

Structural evolution is a promising yet challenging direction in morphology-behavior co-optimization. In this subsection, we provide a detailed discussion of our decisions, experimental findings, and future directions regarding structural evolution.

**Evolving from Scratch and Predefined Morphology Structures.**    For the Ant Navigation task, we chose to evolve morphology skeletons from scratch, starting from the "atomic morphology" shown in Figure 1. This setup represents the most challenging scenario in co-evolution, allowing us to demonstrate the effectiveness of our method while avoiding biases introduced by predefined morphologies. In contrast, the Ants Sumo and Humanoid Navigation tasks use predefined morphology structures, where only attribute development is allowed. This decision are driven by the challenge of defining clear boundaries for structural transformations across different starting morphologies, particularly for complex predefined morphology structures.

**Challenges of Structual Evolution.**    Structural evolution is inherently complex and remains underexplored. Several key issues must be addressed, such as how many body parts can grow from a torso node, how many can branch from each node, which nodes require constraints, the maximum structural depth, and the corresponding attribute transformations. Without carefully designed constraints, the search space grows exponentially, making optimization inefficient and unstable.

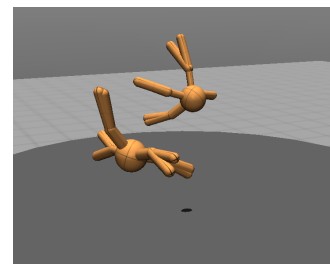

Figure 13: EvoAnts Sumo.

**Structual Evolution for Ants Sumo.**    In the EvoAnts Sumo task (see Figure 13), structural evolution can lead to uncoordinated development of multiple body parts, resulting in excessive forces that cause the agent to fling itself out of the arena. This instability prevents meaningful results, which is why we only included the outcomes of attribute evolution for this task.

**Structural Evolution for Humanoids.**    Training humanoids is significantly more challenging compared to simpler morphologies like the ant. Achieving stable behavior in attribute evolution already requires extensive training and computational resources. Adding structural evolution to the mix would drastically increase complexity, demanding even longer training times and greater computational resources. Given our current computational limitations, we are unable to conduct experiments involving structural evolution for humanoids. Nonetheless, we acknowledge the potential value of this research direction and will pursue it in future work.

**The Complexity of 3D Structural Evolution.**   Incorporating structural evolution into 3D tasks is exceptionally challenging. The interplay between the dimensionality of the environment, task, and action space—including morphology structure, attributes, and behaviors—creates a compounded effect that leads to an exponential explosion in the optimization search space. Furthermore, such a problem setup has not been systematically explored before. Prior work has typically focused on 2D spaces (Ha, 2019), 3D spaces with 2D structural tasks (Yuan et al., 2022; Gupta et al., 2021; Dong et al., 2023), tasks that do not incorporate structural evolution (Huang et al., 2024), or have imposed rigid constraints on morphology, such as bilateral symmetry (Dong et al., 2023), limiting the natural development of morphology through interaction with the environment.

Our work represents a crucial step forward by leveraging geometric symmetries to reduce this complexity. We believe our method lays the foundation for exploring this valuable and fascinating problem setup. While our current results are a small step, they highlight the necessity and significance of addressing these challenges to advance the field further.

**Future Directions.**   The limitations and insights gained from our experiments point to several exciting future directions. These include extending structural evolution to more complex morphologies like humanoids, refining structural constraints to balance exploration and optimization, and systematically investigating the role of geometric symmetries in 3D structural tasks. By continuing to explore these directions, we aim to address the challenges of structural evolution and unlock its full potential for embodied intelligence and robotic design.

Table 2: Hyperparameter Settings for EquiEvo

| Hyperparameter | Setting |
|---|---|
| Number of structural transformations $N_s$ | 5 |
| Number of attribute transformations $N_a$ | 1 |
| Number of control executions $N_e$ | 1,000 |
| Topological graph neural network layer type | GraphConv |
| Geometric graph neural network layer type | SubequivariantMP |
| Topological graph neural network size (structural transformation phase) | (64, 64, 64) |
| Topological graph neural network size (attribute transformation phase) | (64, 64, 64) |
| Geometric graph neural network vector size (geometric subgroup equivariant transformation phase) | (16, 16, 16) |
| Geometric graph neural network scalar feature size (geometric subgroup equivariant transformation phase) | (32, 32, 32) |
| Topological graph neural network size (control execution phase) | (64, 64, 64) |
| Policy learning rate | 5e-5 |
| Value estimation graph neural network size | (64, 64, 64) |
| Value estimation MLP size | (512, 256) |
| Value learning rate | 3e-4 |
| PPO clipping $\epsilon$ | 0.2 |
| PPO batch size | 50000 |
| PPO mini-batch size | 2048 |
| PPO iterations per batch | 10 |
| Training epochs | 1000 |
| Discount factor $\gamma$ | 0.995 |
| GAE $\lambda$ | 0.95 |

---

**Algorithm 1** EquiEvo

---

1: Initialize policy $\pi_\theta$ and value $V_\phi$ {including topological graph neural networks $\varphi_\theta^{\text{s}}$, $\varphi_\theta^{\text{a}}$, $\varphi_\theta^{\text{b}}$, $\varphi_\phi$, and subequivariant graph neural network $\varphi_\theta^{\text{g}}$}
2: **while** not reached maximum iteration **do**
3:    Initialize memory $\mathbb{M} \leftarrow \emptyset$
4:    **while** $\mathbb{M}$ not reached batch size **do**
5:       $\mathcal{G}^{\text{m}} \leftarrow$ initial morphology topology graph
      {Structure Transform Stage}
6:       **for** $t = 0, 1, \ldots, N_{\text{s}} - 1$ **do**
7:          Sample structure transform action $a_t^{\text{s}} \sim \varphi_\theta^{\text{s}}(\mathcal{G}^{\text{m}})$
8:          $\Phi \leftarrow$ Structure
9:          $\mathcal{G}^{\text{m}} \leftarrow$ apply $\boldsymbol{a}^{\text{s}}$ to modify $\mathcal{G}^{\text{m}}$'s morphology structure $(\mathcal{V}, \mathcal{E})$
10:          $r \leftarrow 0$; Store $(r, \boldsymbol{a}^{\text{s}}, \mathcal{G}^{\text{m}}, \Phi)$ to $\mathbb{M}$
11:       **end for**
      {Attribute Transform Stage}
12:       **for** $t = N_{\text{s}}, \ldots, N_{\text{s}} + N_{\text{a}} - 1$ **do**
13:          Sample attribute transform action $\boldsymbol{a}^{\text{a}} \sim \varphi_\theta^{\text{a}}(\mathcal{G}^{\text{m}})$
14:          $\Phi \leftarrow$ Attribute
15:          $\mathcal{G}^{\text{m}} \leftarrow$ apply $a^{\text{a}}$ to modify $\mathcal{G}^{\text{m}}$'s bone joint attributes $\boldsymbol{H}^{\text{m}}$
16:          $r \leftarrow 0$; Store $(r, \boldsymbol{a}^{\text{a}}, \mathcal{G}^{\text{m}}, \Phi)$ to $\mathbb{M}$
17:       **end for**
      {Behavior Control Stage}
18:       $\vec{\mathcal{G}}_{\vec{\boldsymbol{g}}, N_{\text{s}} + N_{\text{a}}} \leftarrow$ initial morphology subequivariant graph
19:       **for** $t = N_{\text{s}} + N_{\text{a}}, \ldots, N_{\text{s}} + N_{\text{a}} + N_{\text{e}} - 1$ **do**
20:          $\vec{\boldsymbol{v}}_t \leftarrow \varphi_\theta^{\text{g}}(\vec{\mathcal{G}}_{\vec{\boldsymbol{g}}})$ {LRF Canonicalization}
21:          Use the calculated LRF transformation vector $\vec{\boldsymbol{u}}, \vec{\boldsymbol{v}}$ to predict a local reference frame and project the morphology subequivariant graph $\vec{\mathcal{G}}_{\vec{\boldsymbol{g}}}$ to the geometry-invariant representation $\mathcal{G}_{\vec{\boldsymbol{g}}}$
22:          Sample motor control action $\boldsymbol{a}^{\text{b}} \sim \varphi_\theta^{\text{b}}(\mathcal{G}_{\vec{\boldsymbol{g}}})$
23:          $\Phi \leftarrow$ Execution
24:          $\mathcal{G}_{\vec{\boldsymbol{g}}} \leftarrow$ environment dynamics $P^{\text{b}}(\mathcal{G}_{\vec{\boldsymbol{g}}}|\mathcal{G}_{\vec{\boldsymbol{g}}}, \boldsymbol{a}^{\text{b}})$
25:          $r \leftarrow$ environment reward; Store $(r, \boldsymbol{a}^{\text{b}}, \mathcal{G}_{\vec{\boldsymbol{g}}}, \Phi)$ to $\mathbb{M}$
26:       **end for**
27:    **end while**{Policy Network Update}
28:    Update policy $\pi_\theta$ and value $V_\phi$ using PPO algorithm based on samples in $\mathbb{M}$
29: **end while**
30: **return** $\pi_\theta$

---

