# OpenReview forum: "Subequivariant Morphology-Behavior Co-Evolution in 3D Environments"
_ICLR.cc/2025/Conference — Submitted to ICLR 2025_

### Official Review · Reviewer_3UE6 · 2024-10-22

**Soundness:** 3
**Presentation:** 3
**Contribution:** 3
**Rating:** 8
**Confidence:** 1

**Summary:**

Sorry I know nothing about this field so I can't give a reasonable review. please just ignore my review.

**Strengths:**

- The paper is well-written and easy to follow.

**Weaknesses:**

none

**Questions:**

none

---

> ### Author Response · Authors · 2024-11-19
> **Response to Reviewer 3UE6**
>
> We sincerely appreciate the reviewer’s kindness and honesty in acknowledging their unfamiliarity with the field, yet still providing a fair and positive assessment of our work. Thank you for your time and consideration in evaluating our paper, and for bringing this to the attention of the area chairs.

---

> > ### Comment · Reviewer_3UE6 · 2024-11-19
> >
> > Thanks for your reply.

---

> > > ### Author Response · Authors · 2024-11-19
> > > **Thank you for raising the score!**
> > >
> > > Dear Reviewer 3UE6,
> > >
> > > We're truly grateful for your feedback, which greatly encourages us! Thank you again for your support!
> > >
> > > Best wishes,
> > >
> > > Authors

---

### Official Review · Reviewer_5q96 · 2024-10-25

**Soundness:** 3
**Presentation:** 3
**Contribution:** 3
**Rating:** 6
**Confidence:** 4

**Summary:**

This work introduces Subequivariant Morphology-Behavior Co-Evolution in 3D Environments (3DS-MB). It further proposes a novel co-evolution approach named EquiEvo, which exploits the geometric symmetries within the proposed task environment to better navigate directional complexities and ensure invariant morphological evolution. Specifically, a local reference frame (LRF) is first predicted from original observations via subequivariant message passing mechanisms, and then the observations are projected into this LRF coordinate so that they become invariant. The authors validate the indispensability of subequivariance modeling for morphology-behavior co-evolution in complex 3D environments. The authors make non-trivial contributions, as they both propose to characterize the directional complexities in 3D environments and present an efficient and generalizable co-evolution framework to tackle this challenge.

**Strengths:**

1.This paper is concisely written and generally easy to follow. The authors clearly state their motivation, including (1) existing 3D benchmark environments lack a reflection of complex spatial geometric structures, and (2) existing co-evolution approaches insufficiently exploit the symmetries in these structures, hence greatly helping with the reader’s understanding of their contributions.

2.The authors present comprehensive experimental results. They not only demonstrate the effectiveness of subequivariance modeling, but also investigate its impact on different model components (such as actor and critic) and compare against hand-crafted normalization methods. The results regarding morphology-task mapping and morphology-behavior mapping are also quite interesting.

3.I appreciate that the authors provide detailed proofs of the invariance of LRF canonicalization and EquiEvo.

**Weaknesses:**

1.The paper implicitly assumes some prior knowledge of subequivariant graph neural networks on the reader’s side. Some of the mathematical notations and formulae (such as the stack of equations on page 5) might need some further clarification so that they are more understandable.

2.I find that the authors did not mention the architectures of the morphology transform policy and “conventional neural networks” involved in behavior control and value estimation. The exact process of morphology transform (such as the actions taken and attributes being transformed) is not explained either. If these design choices are identical to some previous work, I suggest that the authors cite the related papers so that readers could refer to them for more details.

**Questions:**

1.Regarding Figure 5, is it that each pair of agents are trained simultaneously and competing against each other? There are some cases where the win rates of both opponents decrease with the training process. Could you further explain how this should be interpreted?

2.The authors did not mention any publicly accessible code repository in the paper or supplementary material. Do you have any plan to open source your code and environment?

3.The baseline algorithms seem a bit simple, more like ablated versions of your proposed method. I wonder how your RL approach compares with evolutionary algorithms (EAs), such as Neural Graph Evolution, in terms of morphological evolution. From my understanding, the control algorithm that you propose could be readily combined with any of those EAs, serving as inner-loop fitness evaluation. Did you carry out any comparison like this?

Thank you!

---

> ### Author Response · Authors · 2024-11-19
> **Response to Reviewer 5q96**
>
> We sincerely thank the reviewers for their encouraging comments and thoughtful suggestions. We appreciate the opportunity to further improve the clarity and completeness of our work, and we will address each of these points in detail.
>
> > W1: The paper implicitly assumes some prior knowledge of subequivariant graph neural networks on the reader’s side. Some of the mathematical notations and formulae (such as the stack of equations on page 5) might need some further clarification so that they are more understandable.
> >
>
> Thank you for your suggestion. We will provide further clarification on the notations and formulae to improve readability.
>
> > W2: I find that the authors did not mention the architectures of the morphology transform policy and “conventional neural networks” involved in behavior control and value estimation. The exact process of morphology transform (such as the actions taken and attributes being transformed) is not explained either. If these design choices are identical to some previous work, I suggest that the authors cite the related papers so that readers could refer to them for more details.
> >
>
> Thank you for your feedback. We have detailed the architectures of the morphology transformation policy and the neural networks for behavior control and value estimation in Sec. 4.2 "Implementations"  and Table 2. We have enhanced the Sec. 3.2 with additional explanations and relevant citations for clarity.
>
> > Q1.Regarding Figure 5, is it that each pair of agents are trained simultaneously and competing against each other? There are some cases where the win rates of both opponents decrease with the training process. Could you further explain how this should be interpreted?
> >
>
> Thank you for your question. Yes, in Figure 5, each pair of agents is trained simultaneously and competes against each other. The observed decrease in win rates for both opponents during training can be attributed to the presence of draw outcomes in the sumo environment, as neither agent wins in these cases. We mention the draw bonus in the reward settings for sumo, which accounts for these situations. Thank you for pointing this out. We have revised the paper to include a special note on draw outcomes in the Termination section for the sumo environment, clarifying how draws impact win rates during training.
>
> > Q2.The authors did not mention any publicly accessible code repository in the paper or supplementary material. Do you have any plan to open source your code and environment?
> >
>
> We plan to open-source our code and environment on GitHub upon the paper's acceptance.
>
> > Q3.The baseline algorithms seem a bit simple, more like ablated versions of your proposed method. I wonder how your RL approach compares with evolutionary algorithms (EAs), such as Neural Graph Evolution, in terms of morphological evolution. From my understanding, the control algorithm that you propose could be readily combined with any of those EAs, serving as inner-loop fitness evaluation. Did you carry out any comparison like this?
> >
>
> Thank you for this insightful question. We did consider comparing our method with evolutionary algorithms (EAs), such as Neural Graph Evolution, but ultimately chose not to include them in this study for several reasons. First, our focus is on introducing subequivariance within a reinforcement learning (RL) framework, where symmetry is directly leveraged to improve sample efficiency in continuous control tasks. In contrast, EAs typically focus on population-based search and often require a significantly larger number of samples, which may not align with our emphasis on sample efficiency. Additionally, integrating EAs would shift the focus of our work and complicate the analysis, as EAs follow a different paradigm for morphological evolution. However, we agree that combining our control algorithm with EAs could be an interesting direction for future work and may serve as a robust inner-loop fitness evaluation.

---

> > ### Comment · Reviewer_5q96 · 2024-11-26
> > **Response to authors**
> >
> > Thank you for the reply.

---

> > > ### Author Response · Authors · 2024-11-28
> > > **Thank you for the positive recognition!**
> > >
> > > Dear Reviewer 5q96,
> > >
> > > We sincerely appreciate your thoughtful review and positive recognition of our work. Your comments, particularly on the clarity of our experimental results and the importance of subequivariant modeling, validate the significance of our contributions. We are grateful for your detailed feedback, which has further strengthened our paper.
> > >
> > > Thank you again for your valuable insights and support!
> > >
> > > Best wishes,
> > >
> > > Authors

---

> ### Author Response · Authors · 2024-11-22
> **Looking forward to your discussion feedback**
>
> Dear Reviewer 5q96,
>
> Thank you for your thoughtful and constructive review.  In response, we have made specific revisions to address your concerns, including clarifying notations and formulae and adding implementation details. We look forward to any further comments you might have before the discussion period concludes.
>
> Best regards,
>
> Authors

---

### Official Review · Reviewer_ymVo · 2024-10-30

**Soundness:** 3
**Presentation:** 2
**Contribution:** 3
**Rating:** 6
**Confidence:** 2

**Summary:**

This paper introduces a  benchmark for subequivariant morphology-behavior coevolution and employs  graph neural networks to effectively integrate geometric symmetry. Overall, the approach is generally sound.

**Strengths:**

1.The proposed method is both reasonable and meaningful for the co-evolution of morphology and behavior in 3D space.

2.The authors conduct extensive ablation experiments to validate the contributions of each module.

**Weaknesses:**

1.The main algorithmic contribution of this work is the introduction of geometric symmetry into the formulation of co-evolution. However, geometric symmetry is a widely recognized concept for enhancing the sample efficiency of interactive learning methods, such as reinforcement learning, making the contribution appear modest.

2.The paper claims to address the lack of spatial geometric information consideration in tasks, yet the navigation environments used in the experiments are relatively simple and lack significant geometric complexity. Real-world navigation tasks typically involve obstacles. The reviewers would appreciate seeing ablation studies in navigation scenarios with obstacles to further demonstrate the proposed method's effectiveness.

3.The co-evolution of morphology and behavior is tested on only two robots, a humanoid and a spider. Could the authors conduct experiments with additional robots, such as a half-cheetah?

**Questions:**

1.The importance of co-evolution and its practical applications should be clarified in the introduction, as it currently appears somewhat niche.

2.Figure 1 does not effectively convey the concept of subequivariance, and the depiction of successful and failed scenarios is confusing (e.g., why is the left scenario a failure and the right a success?).

3.Figures 7, 8, and 9 indicate that some methods continue to show improvement even after the training process reaches 100%. Why are results from additional training rounds not presented?

---

> ### Author Response · Authors · 2024-11-19
> **Response to Reviewer ymVo (1/2)**
>
> We sincerely thank the reviewers for their constructive feedback and positive assessment of our work. We appreciate the insights provided and will carefully address each point to improve the clarity and robustness of our study.
>
> > W1.The main algorithmic contribution of this work is the introduction of geometric symmetry into the formulation of co-evolution. However, geometric symmetry is a widely recognized concept for enhancing the sample efficiency of interactive learning methods, such as reinforcement learning, making the contribution appear modest.
> >
>
> We agree that geometric symmetry is a widely recognized concept for improving sample efficiency in interactive learning methods like reinforcement learning. Rather than focusing solely on behavior, we leverage subequivariance in a way that impacts both behavior and morphology. The morphology network’s learning signal is derived from rewards generated through behavior-environment interactions. Through this interaction-driven mechanism, subequivariance indirectly influences morphology evolution, contributing to an integrated framework for co-evolution.
>
> Furthermore, the evolution of morphology is fundamentally shaped by environmental interactions. The geometric symmetries present in the environment and its dynamics profoundly influence both morphology and behavior. Previous methods have often overlooked environmental symmetries or imposed direct constraints on morphology, such as bilateral symmetry, without allowing for natural, autonomous development through environmental interaction. Our framework introduces an environment that embodies these geometric symmetries, facilitating the autonomous co-evolution of morphology and behavior. This aligns with the principles of embodied intelligence, where agents learn and evolve through interactions with their environment, leading to more natural and effective adaptations.
>
> Please refer to our General Response. We have revised the first paragraph of the introduction, with the modifications highlighted in blue text, and have re-uploaded the PDF to emphasize the importance and challenges of this problem setup.
>
> > W2: The paper claims to address the lack of spatial geometric information consideration in tasks, yet the navigation environments used in the experiments are relatively simple and lack significant geometric complexity. Real-world navigation tasks typically involve obstacles. The reviewers would appreciate seeing ablation studies in navigation scenarios with obstacles to further demonstrate the proposed method's effectiveness.
> >
>
> We appreciate the reviewer's insightful comments regarding the geometric complexity of our navigation environments. In real-world scenarios, navigation tasks often involve obstacles, leading to exponential growth in the system's geometric transformations. To manage this complexity, navigation can be decomposed into simpler, obstacle-free local environments through planning. Our current work focuses on leveraging geometric equivariance to evolve morphologies that enhance behavioral efficiency in such symmetric environments. Exploring how morphology-behavior co-evolution can facilitate planning capabilities in more complex settings is a valuable direction for future research. Notably, certain organisms, such as slime molds, lack a central nervous system yet effectively navigate simple environments through morphology-behavior co-evolution.

---

> > ### Author Response · Authors · 2024-11-19
> > **Response to Reviewer ymVo (2/2)**
> >
> > > W3: The co-evolution of morphology and behavior is tested on only two robots, a humanoid and a spider. Could the authors conduct experiments with additional robots, such as a half-cheetah?
> > >
> >
> > We thank the reviewer for highlighting concerns about symmetry and generalizability across varied morphologies.  Firstly, the half-cheetah's initial morphology is highly unsuitable for omnidirectional movement in a 3D environment.
> >
> > Second, it is technically feasible to start evolving from any morphology and adjust the attributes or structural depth. However, we made the following considerations for skeleton evolution:
> >
> > - Skeleton evolution inherently requires transforming the attributes of newly grown skeletons.
> > - Evolving from scratch represents the most challenging setup in the co-optimization problem, which demonstrates the effectiveness of our method to the greatest extent.
> > - Evolving from different morphologies introduces prior knowledge about the final morphology, which could be considered a form of bias or "cheating."
> >
> > As such, we focused on evolving morphology skeletons from scratch. For other starting morphologies, determining the boundaries for structural transformations becomes ambiguous, so we only considered attribute transformations in our experiments.
> >
> > > Q1: The importance of co-evolution and its practical applications should be clarified in the introduction, as it currently appears somewhat niche.
> > >
> >
> > Thank you for your insightful comment. In our revision, we will include an explanation of the importance of morphology and behavior co-evolution.
> >
> > > Q2: Figure 1 does not effectively convey the concept of subequivariance, and the depiction of successful and failed scenarios is confusing (e.g., why is the left scenario a failure and the right a success?).
> > >
> >
> > Thank you for your feedback regarding Figure 1. We understand that the concept of subequivariance and the distinction between successful and failed scenarios may not be immediately clear. The green sphere represents the navigation goal. The left scenario depicts a failure due to the agent’s loss of balance, which prevents it from reaching the target. In contrast, the right scenario demonstrates success achieved through the subequivariance method, which introduces radially symmetry, enabling the agent to reach the goal efficiently.
> >
> > > Q3: Figures 7, 8, and 9 indicate that some methods continue to show improvement even after the training process reaches 100%. Why are results from additional training rounds not presented?
> > >
> >
> > We appreciate the reviewer’s observation. Leveraging geometric equivariance primarily improves sample efficiency, meaning that with sufficient training, the results will converge with the baseline. Given the high computational cost of reinforcement learning, especially for institutions with limited resources, we limited the number of epochs to a reasonable level that was sufficient to validate the objectives of our experiments.

---

> ### Author Response · Authors · 2024-11-22
> **Looking forward to your discussion feedback**
>
> Dear Reviewer ymVo,
>
> We deeply appreciate your detailed and constructive review. In response to your suggestions, we have made specific revisions, including clarifications in the Introduction regarding the importance of co-evolution, and detailed explanations in the supplementary material. As the discussion deadline approaches, your perspective has already been invaluable in refining our work, and any additional feedback would help us ensure its value.
>
> Best regards,
>
> Authors

---

> > ### Comment · Reviewer_ymVo · 2024-11-25
> > **Official Comment by Reviewer ymVo**
> >
> > I appreciate the authors taking the time to answer my questions, but I do lack knowledge of the field. The editor doesn't have to be particularly concerned with my suggestions.

---

> > > ### Author Response · Authors · 2024-11-25
> > > **Appreciating Your Indispensable Review!**
> > >
> > > Dear Reviewer ymVo,
> > >
> > > Thank you again for your engagement with our work. From your comments, particularly regarding the role of geometric symmetry in reinforcement learning, it is clear that  you have a thoughtful understanding of the challenges and nuances of this field, which are central to the focus of our work.
> > >
> > > > Our work addresses a complex co-evolution problem, as formalized in Equation (1):
> > > >
> > > >
> > > > $$G_{m} = \text{argmax} J(\pi_\mathcal{G}, \mathcal{G}_{m})$$
> > > >
> > > > $$\text{s.t.} \quad \pi_\mathcal{G} = \text{argmax} J(\pi, \mathcal{G}_{\mathrm{m}})$$
> > > >
> > > > where $J(\pi, G_{m})$ represents the cumulative reward for a given morphology $G_{m}$ and policy $\pi$. From this two-layer objective, it is clear that the optimization of morphology depends on the behavior.  It is important to note that the objective $J$  is defined as an expectation, meaning that in the expectation sense.
> > > >
> > > > In practice, the bi-level optimization is implemented as a co-optimization process, where the inner and outer optimizations alternate incrementally rather than fully optimizing inner layer. Since the evaluation of  $J$  relies on **finite samples** of the behavior policy, if the behavior policy is not equivariant, the **limited sampling** will cause $J$  to vary under sampled initial state. Geometric symmetry reduces the variability introduced by sampling, as equivariant networks ensure **equivalent actions and consistent values**, providing **consistent feedback** across state transformations——such as translations, rotations, or reflections.
> > > >
> > > > By incorporating geometric symmetry into our framework, we extend its application beyond **behavior control to the more intricate morphology-behavior co-evolution problem**. Our approach ensures not only sample efficiency but also consistent and robust evolution in complex 3D environments. We believe this underscores the novelty and significance of our approach and its impact on the field.
> > > >
> > >
> > > We sincerely appreciate your review and believe your comments reflect a thoughtful engagement with the topic. Your thoughtful suggestions have helped us refine our work and further clarify its contributions. Thank you for your valuable support and encouragement—it means a great deal to us.
> > >
> > > Best regards,
> > >
> > > Authors

---

### Official Review · Reviewer_KceS · 2024-11-01

**Soundness:** 3
**Presentation:** 2
**Contribution:** 2
**Rating:** 5
**Confidence:** 3

**Summary:**

The paper addresses the problem of learning control policies for embodied agents living in a 3D environment. It proposes a method for the co-evolution of the behavior policy and the morphology of the agent. The authors use subequivariant graphs for the state representation to take advantage of the symmetries of the tasks with respect to certain transformations (e.g., rotation, translation) and reduce the complexity of the problem. They further propose to use subequivariant graphs to represent the agent's morphology. The paper experimentally shows on three different tasks subequivariance and morphology transformations improve performance of the agent.

**Strengths:**

Integrating subequivariant graph neural networks with morphology-behavior co-evolution is an interesting experiment. The experiments show that both morphology transformations and equivariance contribute to the performance improvement over standard graph neural networks.

**Weaknesses:**

The exposition of the motivations, the methods and the results should be improved. I provide some examples. In the summary of the contributions. 3DS-MB is defines as a "benchmark", in the rest of the paper as a "method". I could not find where the authors define which symmetries the network and the morphology are invariant to. From the figures it seems that the ant evolves in a radially-symmetric way (figure 2), but then in figure 10 it seems to be laterally symmetric.

I find the premise of the paper surprising, as the second sentence of the abstract is "While recent studies have highlighted the considerable benefits of geometric symmetry for tasks like learning to locomote, navigate and explore in dynamic 3D environments, its role within co-evolution setup remains unexplored". Then the paper cites, in the first sentence of the introduction, Gupta et al. [1] and Dong et al. [2], which both describe an evolutionary experiment simultaneously improving the agent's policy and morphology *imposing bilateral symmetry of the agent* (cf. Unimal designed described in the paper). Maybe the authors mean that the studying the role of symmetry was not the focus of the paper, but their wording suggests something else. I recommend to clarify how the author's approach to symmetry in co-evolution differs from or extends beyond the bilateral symmetry used in the cited papers, and if the symmetry they impose is only in the state representation or also in the morphology transformations.

It is unclear when reading the paper which components of the design are novel and which have been taken from previous work. For example, in the section "Morphology transform", no previous paper is mentioned, but several papers cited in the introduction define similar or identical transformations. Instead, the authors write "We divide the morphology transform stage into two sub-stages", implying that is an idea of this paper. In general, it is very difficult to evaluate this paper because the design choices are not introduced in view of the related works. Similarly, how does the subequivariant graph neural network described in this paper differ from the ones used in [3] and [4]?

When I first read the paper, I misunderstood a central aspect of the paper. In the introduction, the authors write "An essential aspect of our approach is ensuring that the evolution of the morphology remains invariant to geometric transformations". At this point, one would expect the morphology transformations to be constrained by some kind of symmetry. Figure 1 and 2 also reinforce this misunderstanding: the morphology on the right (with subequivariance) looks symmetric (radially and laterally), while the one on the left is asymmetric. Instead, it seems to be that, while the representation is symmetric due to the subequivariance of the graph, the morphology transformations are not (figure 10). The authors should make these aspects clearer.

The experimental results do not compare the methods with previously introduced algorithms, they are rather ablation studies. They remove subequivariance, morphology transformations or both. However, the authors mention several previous works about morphology-policy co-evolution. As the initial motivation of the paper is that these methods underperform in 3D environments, they should prove this with experiments.

Overall, while the results are promising, unfortunately I think the paper is not ready for publication in its current state.


[1] Agrim Gupta, Silvio Savarese, Surya Ganguli, and Li Fei-Fei. Embodied intelligence via learning
and evolution. Nature communications, 12(1):5721, 2021.
[2] Heng Dong, Junyu Zhang, Tonghan Wang, and Chongjie Zhang. Symmetry-aware robot design with
structured subgroups. In International Conference on Machine Learning
[3] Runfa Chen, ling Wang, Yu Du, Tianrui Xue, Fuchun Sun, Jianwei Zhang, and Wenbing Huang.
Subequivariant reinforcement learning in 3d multi-entity physical environments. In International
Conference on Machine Learning. PMLR, 2024.
[4] Runfa Chen, ling Wang, Yu Du, Tianrui Xue, Fuchun Sun, Jianwei Zhang, and Wenbing Huang.
Subequivariant reinforcement learning in 3d multi-entity physical environments. In International
Conference on Machine Learning. PMLR, 2024.

**Questions:**

* When the authors speak about "Morphology Value", do they mean a value function mapping the morphology to the expected cumulative reward? I agree that this should not depend on transformations of the environment such as rotations or translations, but a value of the morphology should not depend on the environment state in the first place. Or do the authors want to learn a value function for a specific morphology when the agent is in a given state? This part needs to be better explained.

* Is the "Local Reference Frame Canonicalization" a novelty of this work? No reference is given in that section, as well as no explanation as to why this canonicalization is performed.

* In the humanoid, why are only the attributes of the body's segments modified, and not the morphology? I believe the morphology changes would not work with the humanoid and I believe the authors performed this experiment, but omitted it from the paper.

* Can you explain the learning curves of Sumo? Why do they decrease at a certain point? In the text you write "The results emphasize the critical impact of subequivariance, which proved more influential than morphology evolution alone in enhancing performance". However, in the figure you only show two learning curves, Ant and EvoAnt.

---

> ### Author Response · Authors · 2024-11-19
> **Response to Reviewer KceS (1/4)**
>
> We sincerely appreciate the reviewer's constructive feedback and thoughtful suggestions, which have greatly helped us improve the clarity and presentation of our paper. We are committed to addressing each of the concerns raised in detail and have incorporated relevant clarifications and citations into our revised version. Below, we provide detailed responses to each point to further highlight the motivations and contributions of our work.
>
> > W1: The exposition of the motivations, the methods and the results should be improved. I provide some examples.
> >
> >
> > In the summary of the contributions. 3DS-MB is defined as a "benchmark", in the rest of the paper as a "method".
> >
> > I could not find where the authors define which symmetries the network and the morphology are invariant to.
> >
> > From the figures it seems that the ant evolves in a radially-symmetric way (figure 2), but then in figure 10 it seems to be laterally symmetric.
> >
>
> > W4: In the introduction, the authors write "An essential aspect of our approach is ensuring that the evolution of the morphology remains invariant to geometric transformations". At this point, one would expect the morphology transformations to be constrained by some kind of symmetry.
> >
> >
> > Figure 1 and 2 also reinforce this misunderstanding: the morphology on the right (with subequivariance) looks symmetric (radially and laterally), while the one on the left is asymmetric. Instead, it seems to be that, while the representation is symmetric due to the subequivariance of the graph, the morphology transformations are not (figure 10). The authors should make these aspects clearer.
> >
>
> We appreciate the reviewer’s suggestion regarding consistency in terminology. In our work, "3DS-MB" refers to the proposed benchmark setup, whereas "EquiEvo" denotes our specific method. To improve clarity, we have updated the title of the methods section, "Setup and method", to clearly distinguish between the benchmark setup and the specific method.
>
> - Regarding the symmetries in morphology and network invariance, we did not impose specific symmetry constraints on the morphology itself. Instead, we applied equivariance constraints to the network to leverage symmetry in the state representation.  In this way, we ensure that the network’s output is equivariant to transformations applied to the input. We described our Eg(3)-equivariant setup in Section 3.1.
> - The evolved morphology is determined by both the objectives of the task and the effectiveness of the optimization. Figures 1 and 2 show the evolved results for the navigation task. In Ant Navigation task setting, radially symmetric morphologies are more effective for omnidirectional movement. With the use of equivariant networks, our method successfully evolved a radially symmetric morphology, while the baseline without equivariant networks evolved an asymmetric morphology due to lower optimization efficiency.
> - In Figure 10(b) and (c), we compare the evolution processes under different task (reward) settings. For the task with a forward reward, the evolved morphology exhibits a laterally symmetric structure with stronger front legs and weaker hind legs, while the task without a forward reward leads to a radially symmetric morphology. This shows that the evolution of morphology is fundamentally shaped by environmental interactions and task demands, rather than being a predefined goal. The comparison between Figures 10(a) and (b) shows that methods without integrating geometric equivariance, due to lower sample efficiency, fail to fully evolve a radially symmetric morphology that meets task requirements. This highlights the necessity of considering geometric equivariance in the co-evolution of morphology and behavior frameworks.

---

> ### Author Response · Authors · 2024-11-19
> **Response to Reviewer KceS (2/4)**
>
> > W2: Maybe the authors mean that the studying the role of symmetry was not the focus of the paper, but their wording suggests something else. I recommend to clarify how the author's approach to symmetry in co-evolution differs from or extends beyond the bilateral symmetry used in the cited papers, and if the symmetry they impose is only in the state representation or also in the morphology transformations.
> >
>
> We appreciate the reviewer's query about symmetry. In our work, we address the co-evolution of morphology and behavior by incorporating the geometric symmetries inherent in real-world dynamics. The evolution of morphology is fundamentally shaped by environmental interactions and task demands. Previous methods have often either overlooked the role of geometric symmetries of the environment, resulting in reduced effectiveness, or imposed rigid constraints on morphology, such as bilateral symmetry, which restricts the natural development of morphology through interaction with the environment. Our framework introduces a novel problem setup that embeds geometric symmetries, facilitating the autonomous co-evolution of morphology and behavior. This aligns with the principles of embodied intelligence, where agents learn and evolve through interactions with their environment, leading to more natural and effective adaptations.
>
> Furthermore, geometric symmetries present in the environment and its dynamics profoundly influence both morphology and behavior. Rather than focusing solely on behavior, we leverage subequivariance in a way that impacts both behavior and morphology. The morphology network’s learning signal is derived from rewards generated through behavior-environment interactions. Through this interaction-driven mechanism, subequivariance indirectly influences morphology evolution, contributing to an integrated framework for co-evolution.
>
> Please refer to our General Response. We have revised the first paragraph of the introduction, with the modifications highlighted in blue text, and have re-uploaded the PDF to emphasize the importance and challenges of this problem setup.
>
> > W3: It is unclear when reading the paper which components of the design are novel and which have been taken from previous work.
> >
> >
> > For example, in the section "Morphology transform", no previous paper is mentioned, but several papers cited in the introduction define similar or identical transformations. Instead, the authors write "We divide the morphology transform stage into two sub-stages", implying that is an idea of this paper.
> >
> > In general, it is very difficult to evaluate this paper because the design choices are not introduced in view of the related works. Similarly, how does the subequivariant graph neural network described in this paper differ from the ones used in [3] and [4]?
> >
>
> Thank you to the reviewer for the insightful feedback. We will carefully review our citations and ensure that prior work is properly referenced to clarify our novel contributions.
>
> - Morphology Transformation: For the morphology transformation stage, we adopted the two-stage design introduced in *Transform2Act*. A core aspect of our approach is that morphology evolution is inherently invariant to spatial transformation, which led us to incorporate an equivariant network to achieve invariance in morphology evolution and equivariance in behavior control, improving the efficiency of the overall co-optimization process.
> - Subequivariant Graph Neural Network: Both our approach and [3][4] implement equivariance based on the core function described in Equation (2). However, there are key differences in the design and application. [3] employs a transformer architecture with a fully connected topology, whereas both [4] and our approach utilize GNNs. The main distinction lies in the graph construction: [4] models multi-agent relationships, requiring an entity assignment to define the graph's topology, while our method focuses on single-agent morphology evolution, directly using the morphology's inherent topology to build the graph. This design aligns with our goal of co-optimizing morphology and behavior efficiently within a single-agent framework. We have already addressed these differences in the Related Work section.

---

> > ### Author Response · Authors · 2024-11-19
> > **Response to Reviewer KceS  (3/4)**
> >
> > > W5: The experimental results do not compare the methods with previously introduced algorithms, they are rather ablation studies. They remove subequivariance, morphology transformations or both. However, the authors mention several previous works about morphology-policy co-evolution. As the initial motivation of the paper is that these methods underperform in 3D environments, they should prove this with experiments.
> > >
> >
> > We will clarify the baselines and method setups in the experiment section. We have selected *Transform2Act* and *CompetEvo* as baselines, representing state-of-the-art approaches for structure transform evolution and attribute transform evolution, respectively. In our implementation, methods without any prefix refer to these baselines—e.g., the Ant model in the navigation task corresponds to *Transform2Act*, while the Humanoid model corresponds to *CompetEvo*. Our approach can be viewed as a plugin that introduces equivariance constraints on the state representation within the network.
> >
> > > Q1: When the authors speak about "Morphology Value", do they mean a value function mapping the morphology to the expected cumulative reward? I agree that this should not depend on transformations of the environment such as rotations or translations, but a value of the morphology should not depend on the environment state in the first place. Or do the authors want to learn a value function for a specific morphology when the agent is in a given state? This part needs to be better explained.
> > >
> >
> > In our co-optimization framework, the quality of a morphology is captured by the *morphology value*, defined as the expected cumulative reward for a given morphology, initial state distribution, and policy $\pi$. For a fixed initial state distribution, the morphology value is independent of environment states such as position or velocity, but its updates still rely on feedback from agent-environment interactions. This feedback has a key property: for  environment states and actions that are equivariant, such as rotated or translated configurations, the feedback should be naturally invariant. By integrating subequivariant neural networks into our co-evolution framework, we ensure consistent feedback for morphology value updates, eliminating the need for extensive sampling to approximate invariance.
> >
> > > Q2: Is the "Local Reference Frame Canonicalization" a novelty of this work? No reference is given in that section, as well as no explanation as to why this canonicalization is performed.
> > >
> >
> > Thank you for pointing this out; we will add appropriate references. According to Han et al. (2024), the Local Reference Frame (LRF) Canonicalization is a common technique in geometric graph neural networks to achieve equivariance (or invariance). The LRF predicts an equivariant local coordinate frame and projects the state into this frame. Since both the state and the LRF are equivariant, the projected state becomes invariant. We chose to use the LRF in our work for its simplicity and effectiveness.
> >
> > ```bash
> > Jiaqi Han, Jiacheng Cen, Liming Wu, Zongzhao Li, Xiangzhe Kong, Rui Jiao, Ziyang Yu, Tingyang Xu, Fandi Wu, Zihe Wang, et al. A survey of geometric graph neural networks: Data structures, models and applications. arXiv preprint arXiv:2403.00485, 2024
> > ```

---

> > > ### Author Response · Authors · 2024-11-19
> > > **Response to Reviewer KceS (4/4)**
> > >
> > > > Q3: In the humanoid, why are only the attributes of the body's segments modified, and not the morphology? I believe the morphology changes would not work with the humanoid and I believe the authors performed this experiment, but omitted it from the paper.
> > > >
> > >
> > > For EvoAnt, we chose to evolve morphology skeletons from scratch to address the most challenging setup and eliminate biases from prior knowledge, starting the Ant Navigation task with a simple ``atomic morphology'' (a torso body).
> > >
> > > It is technically feasible to start evolving from any morphology and adjust the attributes or structural depth. However, we made the following considerations for skeleton evolution:
> > >
> > > - Skeleton evolution inherently requires transforming the attributes of newly grown skeletons.
> > > - Evolving from scratch represents the most challenging setup in the co-optimization problem, which demonstrates the effectiveness of our method to the greatest extent.
> > > - Evolving from different morphologies introduces prior knowledge about the final morphology, which could be considered a form of bias or "cheating."
> > >
> > > As such, we focused on evolving morphology skeletons from scratch. For other starting morphologies, we focused solely on attribute transformations to maintain task-specific constraints, as structural transformation boundaries are less well-defined.
> > >
> > > > Q4: Can you explain the learning curves of Sumo? Why do they decrease at a certain point? In the text you write "The results emphasize the critical impact of subequivariance, which proved more influential than morphology evolution alone in enhancing performance". However, in the figure you only show two learning curves, Ant and EvoAnt.
> > > >
> > >
> > > In Sumo, we trained four types of agents: EquiEvoAnt, EquiAnt, EvoAnt, and Ant. These agents were tested in pairs, resulting in six experimental groups. The results show that EvoAnt and Ant performed similarly, indicating that evolution without equivariance did not yield an effective morphology. EquiEvoAnt outperformed the other three agents, demonstrating the effectiveness of combining equivariance with evolution for performance improvement. Additionally, while EquiAnt outperformed Ant, EquiEvoAnt demonstrated even stronger performance than EquiAnt, indicating that the gains from EquiEvoAnt stem not only from equivariant behavior control policy but also from the enhanced effectiveness of morphology evolution facilitated by equivariance.

---

> > > > ### Comment · Reviewer_KceS · 2024-11-20
> > > >
> > > > I thank the authors for the time they took to throughly answer my questions. Some of my concerns have been addressed (e.g., I the paper indeed compares the method with strong baselines, as no subequivariance is equivalent to the SOTA methods mentioned in the related works). I still have some issues with some of the explanations provided.
> > > >
> > > > *In our co-optimization framework, the quality of a morphology is captured by the morphology value, defined as the expected cumulative reward for a given morphology, initial state distribution, and policy . For a fixed initial state distribution, the morphology value is independent of environment states such as position or velocity, but its updates still rely on feedback from agent-environment interactions. This feedback has a key property: for environment states and actions that are equivariant, such as rotated or translated configurations, the feedback should be naturally invariant. By integrating subequivariant neural networks into our co-evolution framework, we ensure consistent feedback for morphology value updates, eliminating the need for extensive sampling to approximate invariance.*
> > > >
> > > > Could the authors clarify whether the morphology value is a function of the state or not? Does it take as input only the structural features of the morphology (e.g., weight, thickness, ...) or also the initial/current pose? The authors say that it depends on the "initial state distribution" - how does the initial state distribution change for a given morphology? Or do you rather mean "the initial state"? I maintain that the value of a morphology should not depend on something that it is not one of its structural properties, and therefore I do not see why a subequivariant representation of the state would make a difference.
> > > >
> > > > Maybe the authors are implying that, as the agent's behavior is the same in every direction, it is easier to learn a value for the morphology, because the task basically becomes 1d instead of 2d/3d, reducing its complexity. In this case I would agree, but if this is the point, I find the explanation overly complicated.
> > > >
> > > > *For EvoAnt, we chose to evolve morphology skeletons from scratch to address the most challenging setup and eliminate biases from prior knowledge, starting the Ant Navigation task with a simple ``atomic morphology'' (a torso body).*
> > > >
> > > > From your figure it seems like you started from a torso and 4 legs. Or did you just start from the sphere? And by coincidence it evolved into a quadruped? I think that at least you are limiting where new limbs can emerge, because I cannot believe that by coincidence they appear just where previous joints are.
> > > >
> > > > *It is technically feasible to start evolving from any morphology and adjust the attributes or structural depth. However, we made the following considerations for skeleton evolution:*
> > > >
> > > > - *Skeleton evolution inherently requires transforming the attributes of newly grown skeletons.*
> > > >
> > > >  - *Evolving from scratch represents the most challenging setup in the co-optimization problem, which demonstrates the effectiveness of our method to the greatest extent.*
> > > >
> > > > - *Evolving from different morphologies introduces prior knowledge about the final morphology, which could be considered a form of bias or "cheating."*
> > > > *As such, we focused on evolving morphology skeletons from scratch. For other starting morphologies, we focused solely on attribute transformations to maintain task-specific constraints, as structural transformation boundaries are less well-defined.*
> > > >
> > > > More than cheating, I would consider it a very interesting experiment to start from a given morphology and fine-tune it to the task. Indeed, you did something similar when you decided to modify the physical properties of the humanoid body. Why not the morphology? I believe that random evolution of the morphology would work well with a dynamically stable body like the humanoid, where removing the wrong segment or adding one in the wrong position might just prevent it from stabilizing. Did the author really decided a-priori not to run morphology evolution on the humanoid, or did they observe that it does not work for that body? Also, in figure 3 the authors show that the Ant morphology actually changes, so in that case it seems that they did apply morphology transformations without starting from scratch. Or am I missing something?

---

> ### Author Response · Authors · 2024-11-20
> **Thanks again for your constructive comments! (1/2)**
>
> Thanks again for your constructive comments, which has significantly contributed to enhancing our paper!
>
> > I1: Could the authors clarify whether the morphology value is a function of the state or not? Does it take as input only the structural features of the morphology (e.g., weight, thickness, ...) or also the initial/current pose? The authors say that it depends on the "initial state distribution" - how does the initial state distribution change for a given morphology? Or do you rather mean "the initial state"? I maintain that the value of a morphology should not depend on something that it is not one of its structural properties, and therefore I do not see why a subequivariant representation of the state would make a difference.
> >
>
> Thank you for your insightful question. The objective in Equation (1) is given by:
>
> $G_{m} = \text{argmax} J(\pi_\mathcal{G}, \mathcal{G}_{m})$
>
> $\text{s.t.} \quad \pi_\mathcal{G} = \text{argmax} J(\pi, \mathcal{G}_{\mathrm{m}}).$
>
> where $J(\pi, G_{m})$ represents the cumulative reward for a given morphology $G_{m}$ and policy $\pi$. From this two-layer objective, it is clear that the optimization of morphology depends on the behavior.  It is important to note that the objective $J$  is defined as an expectation, meaning that in the expectation sense, all states are integrated out. Therefore, the morphology’s policy and value functions ultimately **depend only on the morphology $G_{m}$ and the behavior policy $\pi_G$.**
>
> In practice, the bi-level optimization is implemented as a co-optimization process, where the inner and outer optimizations alternate incrementally rather than fully optimizing inner layer. Since the evaluation of  $J$  relies on finite samples of the behavior policy, if the behavior policy is not equivariant, the limited sampling will cause $J$  to vary under sampled initial state. Geometric symmetry reduces the variability introduced by sampling, as equivariant networks ensure equivalent actions and consistent values, providing consistent feedback across state transformations——such as translations, rotations, or reflections.
>
> Besides, the "initial state distribution" refers to the set of possible initial configurations for the environment and the agent at the start of a task, such as the initial pose, position, or orientation of the agent, as well as any task-specific initialization parameters. This distribution is fixed for a given task. For example, in navigation tasks, the initial state distribution might include random positions and orientations of the agent within a defined space. The cumulative reward $J(\pi, \mathcal{G}_{\mathrm{m}})$ is evaluated under this fixed initial state distribution, which implies that morphologies are optimized to generalize across the different initial states.
>
> We apologize for any confusion in our earlier rebuttal and hope this provides a clearer and more intuitive understanding of the relationship between morphology and state.
>
> > I2: Maybe the authors are implying that, as the agent's behavior is the same in every direction, it is easier to learn a value for the morphology, because the task basically becomes 1d instead of 2d/3d, reducing its complexity. In this case I would agree, but if this is the point, I find the explanation overly complicated.
> >
>
> Your explanation is indeed intuitive and aligns closely with the concept of rotational equivariance. Our goal is to utilize geometric equivariance of transformations such as translations, rotations, and reflections in Euclidean space to enhance the efficiency of co-evolution. This approach corresponds to what  van der Pol et al. (2020)  described as "a reduction in an MDP’s state-action space under an MDP homomorphism."
>
> The complexity of our explanation arises from the nature of the co-evolution process, which is framed as an MDP that involves both policy and value equivariance. Furthermore, the intertwined evolution of morphology and behavior adds another layer of interaction, where the optimization of one impacts the other. We appreciate your perspective and will consider including a simple case similar to your example to provide researchers with a more intuitive explanation.
>
> ```
> Elise van der Pol, Daniel E. Worrall, Herke van Hoof, Frans A. Oliehoek, and Max Welling. MDP homomorphic networks: Group symmetries in reinforcement learning. In  Advances in Neural Information Processing Systems 33, NeurIPS 2020, December 6-12, 2020, virtual, 2020.
> ```

---

> > ### Author Response · Authors · 2024-11-20
> > **Thanks again for your constructive comments! (2/2)**
> >
> > > I3: From your figure it seems like you started from a torso and 4 legs. Or did you just start from the sphere? And by coincidence it evolved into a quadruped? I think that at least you are limiting where new limbs can emerge, because I cannot believe that by coincidence they appear just where previous joints are.
> > >
> >
> > We did impose constraints by uniformly distributing and limiting the growth of legs to four predefined torso positions. We will provide a more detailed explanation of this implementation in the revised manuscript.
> >
> > > I4: More than cheating, I would consider it a very interesting experiment to start from a given morphology and fine-tune it to the task. Indeed, you did something similar when you decided to modify the physical properties of the humanoid body. Why not the morphology? I believe that random evolution of the morphology would work well with a dynamically stable body like the humanoid, where removing the wrong segment or adding one in the wrong position might just prevent it from stabilizing. Did the author really decided a-priori not to run morphology evolution on the humanoid, or did they observe that it does not work for that body?
> > >
> > >
> > > Also, in figure 3 the authors show that the Ant morphology actually changes, so in that case it seems that they did apply morphology transformations without starting from scratch. Or am I missing something?
> > >
> >
> > We appreciate the reviewer's query about structure transform.
> >
> > - First, regarding Figure 3 in Ant Navigation, the morphology evolved from the ``atomic morphology'' shown in Figure 1. In contrast, the Ants Sumo and Humanoid Navigation tasks use predefined morphology structures, allowing only attribute development without structural evolution.
> > - Structural evolution is indeed an interesting and challenging research direction, but it remains underexplored and not yet well-defined. For example, questions such as how many body parts can extend from a torso node, how many can branch from each node, which nodes require constraints, the maximum structural depth, and the corresponding attribute transformations on each node all need to be addressed. Without appropriate constraints, the search space grows exponentially, making optimization highly inefficient.
> > - For simpler morphologies like the ant, structural evolution in complex tasks such as Sumo often leads to unstable or odd behaviors during training. Specifically, when multiple body parts are developed, the lack of proper coordination between them can result in excessive forces, causing the agent to fling itself out of the arena. This instability makes it difficult to achieve meaningful results, which is why we only included the outcomes of attribute evolution for the Sumo task. Exploring structural evolution further will require carefully defined design constraints, allowing for the optimal morphology to emerge through interaction with the environment within a reasonable exploration space.
> > - For humanoids, training is even more challenging. For example, the maximum epoch for training ant behaviors is 500, whereas for humanoids, it is set to 2000. Achieving stable behavior is already difficult, and incorporating both structural evolution and attribute development significantly increases complexity. Increased complexity demands longer training times and substantial computational resources. Due to our current computational limitations, we were unable to conduct experiments involving structural evolution for the humanoid. However, we appreciate your suggestion and will continue to explore these exciting experiments in future work to produce meaningful results.
> > - In 3D spaces, conducting 3D tasks and incorporating morphology structural evolution is an exceptionally challenging endeavor. The interplay between the dimensionality of the environment, the task, and the action space—including morphology structure, attributes, and behaviors—creates a compounded effect, leading to an exponential explosion in the optimization search space. Such a problem setup has not been explored before—prior work has either focused on 2D spaces, 3D spaces with 2D structural tasks, or, even if both criteria are met, has not incorporated morphology structural evolution. We believe our work, which leverages geometric symmetries to reduce this complexity, represents a crucial step forward for the community. While it may be a small step, it is a necessary, significant, and impactful one, laying the foundation for continued exploration of this valuable and fascinating problem setup.
> >
> > We deeply appreciate your detailed and thoughtful review.  Your dedicated effort and insightful feedback have not only inspired us but also reinforced our determination to continue exploring this challenging and meaningful direction.

---

> > > ### Comment · Reviewer_KceS · 2024-11-20
> > >
> > > I thank the authors for the once again detailed answers, I am now more confident about the soundness of the paper and I have increased my score. I would encourage the authors to also include the negative results that they have obtained (or that they will obtain in future experiments), maybe just citing them in the discussion or in the supplementary material, because they can avoid other researchers pursuing these unfruitful directions.

---

> > > > ### Author Response · Authors · 2024-11-21
> > > > **Thank you for the support!**
> > > >
> > > > Dear Reviewer KceS,
> > > >
> > > > Thank you very much for your thoughtful suggestions and encouragement! Following your recommendations, we have incorporated the suggested presentation clarity into the paper, which have helped us improve the paper!  We hope these revisions better illuminate the value of our work. As the discussion deadline nears, we eagerly await any additional comments or questions you may have.
> > > >
> > > > **We appreciate the reviews and thank you again for considering our score thoughtfully!**
> > > >
> > > > **P.S.** We have made the following updates in response to your feedback:
> > > >
> > > > - Added **Section D. The Role of Geometric Symmetry in Morphology Evolution** and **E.4 Discussion on Structural Evolution** (including a demo visualization of **EvoAnt's structural evolution failure in Sumo**, **Figure 12**) in the supplementary material.
> > > > - Included a simple case similar to your example in the last paragraph of the **Introduction** to provide researchers with a more intuitive explanation.
> > > > - Added a citation to **Section D** in the final paragraph of the **Section 3.1 Extension to 3DS-MB** to provide further explanation.
> > > > - Cited Section **E.4** in the **Conclusion** for more discussions.
> > > >
> > > > The updated anonymous paper has been re-uploaded, with all new changes highlighted in **red**.
> > > >
> > > > Best wishes,
> > > >
> > > > The Authors

---

### Official Review · Reviewer_Zyvk · 2024-11-02

**Soundness:** 2
**Presentation:** 2
**Contribution:** 1
**Rating:** 1
**Confidence:** 5

**Summary:**

This submission proposed a strategy for morphology+control co-optimization that incorporates subequivariance-based geometric symmetry into the policy module as an inductive bias to accelerate training in 3D environments. This approach constructs a local orthonormal basis and projects sensing information with spatial properties such as velocity and position into this local basis to enhance the generalizability of the policy module over diverse transformations, particularly those that result from translations, rotations or reflections of the morphology or task.

**Strengths:**

Originality

This work provides an original extension to morphology+control co-optimization frameworks that explicitly address geometric symmetry in policy learning.

Quality and clarity

The authors made an effort to formalize their policy architecture with mathematical definitions, which should help reduce potential misunderstandings when others attempt to reproduce their results.

Significance

Co-optimization of morphology and control is an important longstanding unsolved problem. Supporting a diverse set of morphologies with a more efficient controller would reduce the time needed for designing morphology-specific network architectures. Attention and effort in this area has significance for robotics and other forms of "embodied intelligence".

**Weaknesses:**

Lack of novelty

The primary contribution of this work is a substitution of the reinforcement learning policy module from the Transform2Act approach with an equivariant GNN. Compared to the work cited by authors (Chen et al., 2023 https://arxiv.org/abs/2305.18951) (Chen et al., 2024 https://arxiv.org/abs/2407.12505), this submission appears to be a repackaging of an existing subequivariant reinforcement learning idea with a different problem setup. This submission reuses the same local reference frame (LRF) transformation as in https://arxiv.org/abs/2407.12505 with the same proof process in their appendix. As far as I can tell, the only appreciable difference is the addition of a relatively simple translation to the Og(3) setup in https://arxiv.org/abs/2407.12505 forming Eg(3). A more compelling contribution would include a demonstration of significant differences in results. While the authors claimed their setup can inject symmetry into generated morphological designs, their claim is not well supported.

Lack of evidence of generalizability

Although the symmetry enhancement is the core claim, improved symmetry was only explored within a single basic agent skeleton -- the ant morphology. There was no metric or quantitative analysis to test the consistency of symmetry improvement across varied morphologies, or how the improvement of implementing LRF transformation in the policy module affects the optimization algorithm.
Without visualizations of generated morphologies from additional environments, and without a symmetry-based metric to measure the improvement across different setups, it is difficult to assess the claims of the submission.

Lack of experiment design and analysis

Following up on the above point, the only metric that was considered was reward. In two of the three explored benchmark environments (the humanoid and sumo) the morphology transformation was omitted and only attribute transformations were possible. In this reduced setting, it was unclear how morphology metrics changed over time (e.g. the limb lengths on both sides of the agent) should be readily available but was not extracted and analyzed. However the submission claims “integrating morphology evolution with sub-equivariance yielded a synergistic effect” (Sect. 4.3) which is not supported by the experiments and could simply be the policy network producing better control under LRF transformation that are independent of morphology.

Lack of clarity

The mathematical notation was inconsistent and rather confusing. For example, symbols like “m” are overloaded with different meanings across subscripts (equation 3), superscripts (equation 3), and variables (equation 9, 10, 11) without further clarification of meaning in a verbal format, which undermines the clarity of the theoretical framework. In addition to restructuring the mathematical framework and standardizing notion, explaining the symbols used in equations with natural language, a few words even, would improve clarity immensely -- a symbol table would be ideal.

**Questions:**

1. Can you provide a concrete analysis of the contribution of subequivariant policy in morphology evolution?

2. Can you provide morphology metrics figures over time? Such as the length of limbs, and compute the symmetric error of design by comparing the difference of limb length divided by total limb length?

2. Can you analyze the features of the group of evolved morphologies at each point of the timeline using multiple trials, and explain how your policy module affects evolution?

3. Can you run experiments starting from different morphologies, and introduce varying levels of freedom (e.g. only allow attribute transform, allow attribute transform plus morphology transform with a certain degree, fully allow both) for each starting point?

4. Can you rewrite the equations, especially 7-17, and move unnecessary details to the appendix to clear space for more experiments?

---

> ### Author Response · Authors · 2024-11-19
> **Response to Reviewer Zyvk (1/3)**
>
> We appreciate the reviewer’s detailed feedback and recognition of our work’s originality. We understand that some points may stem from misinterpretations of our methodology. We are preparing responses to clarify these aspects and address the questions raised. Thank you for your comments, which will help us further improve the clarity and rigor of our presentation.
>
> > W1: The primary contribution of this work is a substitution of the reinforcement learning policy module from the Transform2Act approach with an equivariant GNN. Compared to the work cited by authors (Chen et al., 2023; Chen et al., 2024), this submission appears to be a repackaging of an existing subequivariant reinforcement learning idea with a different problem setup.
> >
>
> Thank you for your feedback regarding the novelty of our work. We would like to clarify that adapting existing methods to address significant problem setups is a common and valuable approach in impactful research. For instance, AlphaGo combined MCTS with DRL to tackle the complex game of Go; AlphaFold series utilized existing Transformer and diffusion model  to predict protein folding; GPT-3 scaled up transformer models with larger datasets and parameters for natural language processing tasks; and DALL-E employed transformers for text-to-image generation. These examples demonstrate that applying established techniques to new domains can yield significant advancements by addressing unique challenges within those domains.
>
> - In our work, we address the co-evolution of morphology and behavior by incorporating the geometric symmetries inherent in real-world dynamics. The evolution of morphology is fundamentally shaped by environmental interactions and task demands. Previous methods have often either overlooked the role of geometric symmetries of the environment, resulting in reduced effectiveness, or imposed rigid constraints on morphology, such as bilateral symmetry, which restricts the natural development of morphology through interaction with the environment. Our framework introduces a novel problem setup that embeds geometric symmetries, facilitating the autonomous co-evolution of morphology and behavior. This aligns with the principles of embodied intelligence, where agents learn and evolve through interactions with their environment, leading to more natural and effective adaptations.
> - Furthermore, geometric symmetries present in the environment and its dynamics profoundly influence both morphology and behavior. Rather than focusing solely on behavior, we leverage subequivariance in a way that impacts both behavior and morphology. The morphology network’s learning signal is derived from rewards generated through behavior-environment interactions. Through this interaction-driven mechanism, subequivariance indirectly influences morphology evolution, contributing to an integrated framework for co-evolution.
>
> Please refer to our General Response. We have revised the first paragraph of the introduction, with the modifications highlighted in blue text, and have re-uploaded the PDF to emphasize the importance and challenges of this problem setup.

---

> ### Author Response · Authors · 2024-11-19
> **Response to Reviewer Zyvk (2/3)**
>
> > W2: Although the symmetry enhancement is the core claim, improved symmetry was only explored within a single basic agent skeleton -- the ant morphology. There was no metric or quantitative analysis to test the consistency of symmetry improvement across varied morphologies, or how the improvement of implementing LRF transformation in the policy module affects the optimization algorithm. Without visualizations of generated morphologies from additional environments, and without a symmetry-based metric to measure the improvement across different setups, it is difficult to assess the claims of the submission. ... In two of the three explored benchmark environments (the humanoid and sumo) the morphology transformation was omitted and only attribute transformations were possible.
> >
>
> We thank the reviewer for highlighting concerns about symmetry and generalizability across varied morphologies. Our primary goal is not to enforce symmetry in morphology per se, but rather to inject geometric symmetry into the policy module to leverage the symmetry inherent in real-world environments and dynamic systems. This enables a more efficient co-optimization of morphology and control. The resulting symmetry in morphology is an emergent property, not the intended objective of our approach.
>
> - To evaluate the effectiveness of our method, we provided visualizations (Fig. 10 for Ant, Fig. 6 for Humanoid) to qualitatively analyze the symmetry in evolved morphologies. Quantitatively, we used the “reward obtained by training duration” metric (shown in Figs. 7, 8, and 9) as it directly reflects the effectiveness of morphology-control co-evolution, which aligns with our motivation.
> - We designed the Ant-Navigation task starting from a simple “atomic morphology” (a torso body) as the initial point. This choice allows the agent to autonomously co-optimize morphology and control based on task requirements and environmental interaction, focusing on the effects of injected symmetry. Examining different starting morphologies was not within our scope, as our objective was to understand the impact of geometric symmetry on co-optimization.
> - Regarding different initial skeletons, like prior work such as CompetEvo, we did not focus on skeleton evolution. Instead, we investigated the impact of morphological attributes on task performance. To this end, we further explored two complex tasks, Humanoid-Navigation and Ant-Sumo, to assess the effect of geometric symmetry injection on morphology-control co-optimization.
>
> We have already reiterated our motivation in the introduction to clarify this.
>
>
> > W3: The only metric that was considered was reward. … The submission claims ‘integrating morphology evolution with sub-equivariance yielded a synergistic effect’ … which is not supported by the experiments and could simply be the policy network producing better control under LRF transformation that are independent of morphology
> >
>
> > Q1: Can you provide a concrete analysis of the contribution of subequivariant policy in morphology evolution?
> >
>
> > Q2: Can you provide morphology metrics figures over time? Such as the length of limbs, and compute the symmetric error of design by comparing the difference of limb length divided by total limb length?
> >
>
> Thank you for the thoughtful feedback. We would like to reiterate that our approach focuses on embedding geometric symmetry in the policy module to exploit the inherent symmetry of real-world environments and dynamic systems, thus enhancing the co-optimization of morphology and control. Symmetry in morphology is an outcome of the optimization process rather than an explicit requirement.
>
> - As reward is the primary metric for evaluating sample efficiency, which aligns with our goals, the plotted reward curves in our results clearly demonstrate that the contribution of the subequivariant policy in morphology evolution lies in improving sample efficiency.
> - To provide additional insights, we have included the evolving processes of Ant in Fig. 10 and compared the morphology changes over time using different methods. These observations are intended to demonstrate that the development of morphology is fundamentally shaped by environmental interactions (states) and task demands (reward), rather than being a predefined goal.
>
> We appreciate the reviewer’s suggestion to further analyze the characteristics of the evolved morphologies. We will consider incorporating such analyses to provide readers with additional insights into the results.

---

> > ### Author Response · Authors · 2024-11-19
> > **Response to Reviewer Zyvk (3/3)**
> >
> > > W4: The mathematical notation was inconsistent and rather confusing. … Explaining the symbols used in equations with natural language, a few words even, would improve clarity immensely -- a symbol table would be ideal.
> > >
> >
> > > Q5: Can you rewrite the equations, especially 7-17, and move unnecessary details to the appendix to clear space for more experiments?
> > >
> >
> > In Equation (3), the symbol “m” in `mathrm` format represents morphology. In Equations (9-11), “m” is written in standard math font and represents the message in the graph neural network.
> >
> > We will provide further clarification on the notations and formulae in Equations (7-17) to improve readability. These equations are critical to our method and form the core of the subequivariant framework. While the equations themselves are indispensable, we have moved unnecessary derivations to the appendix to ensure the main content remains clear and accessible for all readers. Please refer to our updated pdf.
> >
> > > Q3: Can you analyze the features of the group of evolved morphologies at each point of the timeline using multiple trials, and explain how your policy module affects evolution?
> > >
> >
> > We appreciate the reviewer’s question. However, such analysis is not applicable to our method as it focuses on co-optimization rather than a population-based approach. We believe the reviewer may be referring to a morphological evolution diagram, such as Figure 3 in NGE (Neural Graph Evolution, https://arxiv.org/pdf/1906.05370). Diagrams like that are specific to population-based methods, where agents evolve as a cluster of generations, allowing for branching based on selection and elimination. In contrast, co-optimization methods follow a single branch (group), which is why this type of comparison is not feasible for our approach.
> >
> > > Q4: Can you run experiments starting from different morphologies, and introduce varying levels of freedom (e.g. only allow attribute transform, allow attribute transform plus morphology transform with a certain degree, fully allow both) for each starting point?
> > >
> >
> > We appreciate the reviewer’s suggestion. Yes, it is technically feasible to start evolving from any morphology and adjust the attributes or structural depth. However, we made the following considerations for skeleton evolution:
> >
> > - Skeleton evolution inherently requires transforming the attributes of newly grown skeletons.
> > - Evolving from scratch represents the most challenging setup in the co-optimization problem, which demonstrates the effectiveness of our method to the greatest extent.
> > - Evolving from different morphologies introduces prior knowledge about the final morphology, which could be considered a form of bias or "cheating."
> >
> > As such, we focused on evolving morphology skeletons from scratch. For other starting morphologies, determining the boundaries for structural transformations becomes ambiguous, so we only considered attribute transformations in our experiments.

---

> ### Author Response · Authors · 2024-11-22
> **Looking forward to your discussion feedback**
>
> Dear Reviewer  Zyvk,
>
> Thank you for your detailed review, which has been crucial in refining our paper. We've made specific revisions to address your concerns, particularly around presentation clarity. With the discussion deadline approaching, your final insights would be invaluable to perfecting our work and would be greatly appreciated.
>
> Best regards,
>
> Authors

---

> > ### Comment · Reviewer_Zyvk · 2024-11-26
> >
> > The mathematical notation has been corrected, improving the paper's clarity. I am raising my score for Presentation to reflect this effort.
> >
> > However, apart from that, the authors reiterate what was in the original submission but do not address the weaknesses I mentioned.
> >
> > > ...such analysis is not applicable to our method as it focuses on co-optimization rather than a population-based approach.
> >
> > Such an analysis can be conducted in the time domain for one experiment with all the designed morphologies under the transform2act framework, or by aggregating results from multiple experiments.

---

> ### Author Response · Authors · 2024-11-28
> **Appreciating Your Encouraging Comments and Suggestions!**
>
> Dear Reviewer Zyvk,
>
> Thank you for taking the time to provide feedback and for raising your score for presentation—we deeply appreciate your recognition of our efforts to improve the clarity of the paper. Your comments and encouragement are invaluable, which have helped us improve the paper!
>
> > Morphological Evolution Analysis.
> >
>
> We acknowledge your suggestion regarding analyzing the features of evolved morphologies over time. To address this, we have incorporated additional visualizations and discussions in the supplementary material to provide more insight into the co-evolution process:
>
> - **Step-by-Step Visualization**: As shown in **Figure 10 (main text)**, we present a step-by-step visualization of the morphology transformation within a single episode. This illustrates the trajectory of morphology transformations during one complete episode of training.
> - **Training (Evolution) Progress Analysis (Supplementary Section D)**: In **Figure 11 (Section D: The Role of Geometric Symmetry in Morphology Evolution)**, we have included a **visualization of morphological evolution across the training process**. This figure compares two settings:
>     - **Policy Without Geometric Symmetry** (left): Morphological evolution appears highly asymmetric and incomplete. Structural transformations often get trapped in local optima early in the process, as reflected in the lower reward curves. This is likely due to the reinforcement learning framework, where abundant exploration is required to optimize both morphology and behavior. In such cases, the interplay between morphology and behavior becomes more prone to local minima, complicating optimization within the vast search space.
>     - **Policy With Geometric Symmetry** (right): The integration of geometric symmetry fosters more intricate and comprehensive morphology evolution, as evidenced by higher reward curves. This improvement supports the claim that leveraging geometric symmetry reduces search space redundancy in a lossless manner, leading to better optimization and higher co-evolution efficiency. Additionally, the training progress demonstrates that **morphological symmetry is not explicitly enforced** but naturally **emerges and varies** throughout the training process, evolving in response to **interactions with the environment and task demands**.
>
> These visualizations aim to highlight how injecting geometric symmetry not only improves reward outcomes but also fosters a more efficient and complete co-evolution of morphology and behavior by better reflecting the symmetry inherent in the environment and task demands, all while addressing the challenging optimization dynamics of reinforcement learning.
>
> **Please review the updated anonymous paper, where updates are highlighted in red, including the caption for Figure 10 and Section D: "The Role of Geometric Symmetry in Morphology Evolution" (featuring Figure 11).** We hope these updates address your concerns about how the policy module influences evolution and offer a more comprehensive view of the evolving morphologies, both within a single episode and throughout different training epochs.
>
> Thank you again for your support! We look forward to any further comments you might have before the discussion period concludes.
>
> Best regards,
>
> Authors

---

### Author Response · Authors · 2024-11-19
**General Response**

We sincerely thank all reviewers and ACs for their time and efforts on reviewing the paper. We are glad that the reviewers recognized the contributions of our paper, which we briefly summarize as follows.

**Originality and Novelty.** *"This work provides an original extension to morphology+control co-optimization frameworks that explicitly address geometric symmetry in policy learning."* (Reviewer Zyvk)  *"Integrating subequivariant graph neural networks with morphology-behavior co-evolution is an interesting experiment."* (Reviewer KceS)

**Relevance and Significance.** *"Co-optimization of morphology and control is an important longstanding unsolved problem."* (Reviewer Zyvk)  *"The proposed method is both reasonable and meaningful for the co-evolution of morphology and behavior in 3D space."* (Reviewer ymVo)

**Thorough Evaluation.** *"The authors conduct extensive ablation experiments to validate the contributions of each module."* (Reviewer ymVo)  *"The authors present comprehensive experimental results. They not only demonstrate the effectiveness of subequivariance modeling, but also investigate its impact on different model components (such as actor and critic) and compare against hand-crafted normalization methods."* (Reviewer 5q96)  *"The results regarding morphology-task mapping and morphology-behavior mapping are also quite interesting."* (Reviewer 5q96)

**Clarity and Rigor.** *"The authors made an effort to formalize their policy architecture with mathematical definitions, which should help reduce potential misunderstandings when others attempt to reproduce their results."* (Reviewer Zyvk)  *"This paper is concisely written and generally easy to follow. The authors clearly state their motivation, including (1) existing 3D benchmark environments lack a reflection of complex spatial geometric structures, and (2) existing co-evolution approaches insufficiently exploit the symmetries in these structures, hence greatly helping with the reader’s understanding of their contributions."* (Reviewer 5q96)  *"I appreciate that the authors provide detailed proofs of the invariance of LRF canonicalization and EquiEvo."* (Reviewer 5q96)

---

> ### Author Response · Authors · 2024-11-19
>
> We also appreciate the reviewers for their thoughtful comments and concerns. Below we summarize three core aspects:
>
> 1. **Main focus and contributions.**
>
>     In our work, we address the co-evolution of morphology and behavior by incorporating the geometric symmetries inherent in real-world dynamics. The evolution of morphology is fundamentally shaped by environmental interactions and task demands. Previous methods have often either overlooked the role of geometric symmetries of the environment, resulting in reduced effectiveness, or imposed rigid constraints on morphology, such as bilateral symmetry, which restricts the natural development of morphology through interaction with the environment. Our framework introduces a novel problem setup that embeds geometric symmetries, facilitating the autonomous co-evolution of morphology and behavior. This aligns with the principles of embodied intelligence, where agents learn and evolve through interactions with their environment, leading to more natural and effective adaptations.
>
>     Furthermore, geometric symmetries present in the environment and its dynamics profoundly influence both morphology and behavior. Rather than focusing solely on behavior, we leverage subequivariance in a way that impacts both behavior and morphology. The morphology network’s learning signal is derived from rewards generated through behavior-environment interactions. Through this interaction-driven mechanism, subequivariance indirectly influences morphology evolution, contributing to an integrated framework for co-evolution.
>
> 2. **What's the symmetry in our paper?**
>
>     Our primary objective is not to enforce symmetry of the morphology but rather to inject geometric symmetry constraints within the policy module, enabling a more efficient co-optimization of morphology and control. Specifically, in our co-optimization framework, the quality of a morphology is captured by the *morphology value*, defined as the expected cumulative reward for a given morphology, initial state distribution, and policy $\pi$. For a fixed initial state distribution, the morphology value is independent of environment states such as position or velocity, but its updates still rely on feedback from agent-environment interactions. This feedback has a key property: for environment states and actions that are equivariant, such as rotated or translated configurations, the feedback should be naturally invariant. By integrating subequivariant neural networks into our co-evolution framework, we ensure consistent feedback for morphology value updates, eliminating the need for extensive sampling to approximate invariance.
>
>     Notably, any symmetric structure observed in the resulting morphology is the outcome of optimization under the given task setting rather than an explicit requirement of our method; if a symmetric morphology arises, it is because such a structure achieves better performance within that specific task.
>
> 3. **How would morphology evolution be affected?**
>
>     The obtained morphology is the outcome of co-optimization, shaped jointly by environmental interactions and task demands. Figures 1 and 2 show the evolved results for the navigation task. In Ant Navigation task setting, radially symmetric morphologies are more effective for omnidirectional movement. With the use of equivariant networks, our method successfully evolved a radially symmetric morphology, while the baseline without equivariant networks evolved an asymmetric morphology due to lower optimization efficiency.
>
>     To provide additional illustration, in Figure 10(b) and (c), we compare the evolution processes under different task (reward) settings. For the task with a forward reward, the evolved morphology exhibits a laterally symmetric structure with stronger front legs and weaker hind legs, while the task without a forward reward leads to a radially symmetric morphology. This shows that the evolution of morphology is fundamentally shaped by environmental interactions and task demands, rather than being a predefined goal.
>
>     The comparison between Figures 10(a) and (b) shows that methods without integrating geometric equivariance, due to lower sample efficiency, fail to fully evolve a radially symmetric morphology that meets task requirements. This highlights the necessity of considering geometric equivariance in the co-evolution of morphology and behavior frameworks.

---

> > ### Author Response · Authors · 2024-11-19
> >
> > - The updated part of introduction is as follows:
> >
> >     **The rich diversity of animal morphologies, shaped over millions of years in complex environments, underscores the deep link between body form and intelligence, where well-adapted morphologies enable agents to learn and perform complex tasks effectively (Gupta et al., 2021; Brooks, 1991). Morphology evolution is fundamentally shaped by environmental interactions and task demands, as supported by evolutionary biology perspectives (Maynard-Smith, 1974; Gould, 2010).** The co-evolution of morphology and behavior in 3D environments has gained significant interest (Sims, 1994; Dong et al., 2023; Chen et al., 2023b; Gupta et al., 2021), focusing on how agents’ form and function evolve together to improve adaptability and performance. **Some works, such as (Gupta et al., 2021; Dong et al., 2023), impose bilateral symmetry constraints directly on morphology, diverging from evolutionary principles, as agents should evolve optimal morphologies through interaction with the environment, rather than relying on predefined constraints.**
> >
> >     **Alternatively, the geometric symmetry of environments and dynamic systems naturally exists, which motivates us to explore how leveraging such symmetry can accelerate the evolution of optimal morphologies through interaction with the environment. Recent studies highlight that leveraging geometric symmetry in tasks like locomotion, navigation, and exploration within dynamic 3D settings can significantly improve the efficiency of behavioral evolution (Chen et al., 2023a; 2024).** While recent co-evolution benchmarks (Yuan et al., 2022; Huang et al., 2024; Liu et al., 2022) have achieved remarkable progress (Liu et al., 2022), they often fail to fully exploit geometric symmetry within 3D spaces. These benchmarks face two main issues: 1) a focus on tasks with limited spatial geometric considerations, such as the  ``move forwar'' objective, which are simplified to fixed directional movements and do not require complex direction-aware strategies; 2) a lack of research exploring the use of geometric symmetry in co-evolution setups to handle the directional complexities of 3D environments effectively.
> >
> >
> > **Please access the updated anonymous paper, with major updates highlighted in blue.**
> >
> > > [1] Agrim Gupta, Silvio Savarese, Surya Ganguli, and Li Fei-Fei. Embodied intelligence via learning and evolution. *Nature communications*, 12(1):5721, 2021.
> > >
> > > [2] Rodney A. Brooks. New approaches to robotics. *Science*, 253(5025):1227–1232, 1991. doi: 10. 1126/science.253.5025.1227.
> > >
> > > [3] John Maynard-Smith. *Models in ecology*. Cambridge university press, 1974.
> > >
> > > [4] Stephen Jay Gould. *The panda’s thumb: More reflections in natural history*. WW Norton & company, 2010.
> > >
> > > [5] Agrim Gupta, Silvio Savarese, Surya Ganguli, and Li Fei-Fei. Embodied intelligence via learning and evolution. *Nature communications*, 12(1):5721, 2021.
> > >
> > > [6] Heng Dong, Junyu Zhang, Tonghan Wang, and Chongjie Zhang. Symmetry-aware robot design with structured subgroups. In *International Conference on Machine Learning*, pp. 8334–8355. PMLR, 2023.
> > >
> > > [7] Runfa Chen, Jiaqi Han, Fuchun Sun, and Wenbing Huang. Subequivariant graph reinforcement learning in 3d environment. In *International Conference on Machine Learning*. PMLR, 2023a.
> > >
> > > [8] Runfa Chen, ling Wang, Yu Du, Tianrui Xue, Fuchun Sun, Jianwei Zhang, and Wenbing Huang. Subequivariant reinforcement learning in 3d multi-entity physical environments. In *International Conference on Machine Learning*. PMLR, 2024.
> > >

---

### Author Response · Authors · 2024-11-21
**General Response to Reviewers' Discussion (1/2)**

We have made several updates in response to the reviewers' discussion to address their valuable feedback and improve the clarity and depth of our paper:

- Added **Section D. The Role of Geometric Symmetry in Morphology Evolution** and **E.4 Discussion on Structural Evolution** in Appendix to provide further insights and address specific concerns.
- Added a citation to **Section D** in the final paragraph of the **Section 3.1 Extension to 3DS-MB** to provide further explanation.
- Cited **Section E.4** in the **Conclusion** to offer additional discussions on structural evolution.

The updated anonymous paper has been **re-uploaded**, with **all new changes** highlighted in **red** for easy reference.

We sincerely thank all reviewers for their constructive feedback, which has greatly helped improve the paper.

---

> ### Author Response · Authors · 2024-11-28
> **General Response to Reviewers' Discussion (2/2)**
>
> In response to the reviewer’s valuable suggestion regarding the analysis of evolved morphologies over time, we have incorporated new visualizations and discussions to provide deeper insights into the co-evolution process. Specifically:
>
> - **Step-by-Step Visualization**: As previously included in the main text (Figure 10), we provide a step-by-step visualization of morphology transformation within a single episode. This visualization already illustrates the trajectory of morphology transformations throughout a complete episode, offering a detailed view of the evolution process. We would like to highlight this existing content to emphasize its role in demonstrating the morphology changes within an episode.
> - **Training Progress Analysis**: In Section D (Supplementary Material), we have introduced Figure 11, which visualizes the morphological evolution across different stages of the training process. This figure compares two settings—one without the integration of geometric symmetry and one with it—highlighting how geometric symmetry fosters more efficient and complete co-evolution. We show how symmetry naturally emerges and evolves over time in response to task demands and environmental interactions, further improving the co-evolution of morphology and behavior.
>
> These updates, particularly **Figure 11**, emphasize the impact of geometric symmetry in facilitating better optimization and more robust morphology evolution, while reflecting the challenges posed by reinforcement learning dynamics.
>
> We hope these new additions help clarify how **the leverage of geometric symmetry via neural networks** influences morphology evolution over time and provide a more comprehensive understanding of the co-evolution process. **The updated anonymous paper**, including these changes, has been re-uploaded with updates **highlighted in red**.
>
> We sincerely thank all reviewers for their insightful comments, which have significantly contributed to the enhancement of this paper.

---

### Meta-Review · Area_Chair_9bVA · 2024-12-20

**Metareview:**

The submission proposed a method for the joint learning ("co-evolution") of the physical morphology of an agent as well as its behavior policy, using a sub-equivariant model. While the initial reviews highlighted strengths such as originality [Zyvk], quality of motivation [Zyvk, ymVo], clarity, [Zyvk, 5q96], demonstrated improvement due to morphology transformations and equivariance [KceS,ymVo, 5q96], detailed proofs of invariance [5q96] as well as addressing an important problem [Zyvk], the majority of the reviews were highly critical of the paper, highlighting multiple weaknesses:

- Novelty,
- Positioning,
- Lack of generalization,
- Problem in the experimental setup, simplicity of the benchmarks,
- missing baselines,
- Writing,

The authors could not provide answers on all of this issues which satisfied the reviewers. Critical weaknesses remained, like novelty, generalization, positioning. The AC judges that this paper is not yet ready for publication. Given the variance in reviewer scores the SAC also reviewed the reviews and resulting discussion and agrees with the decision recommended by the AC.

One of the 5 reviews was completely empty and the high placeholder score of 8 provided was entirely ignored by the AC in the assessment of the paper.

**Additional Comments On Reviewer Discussion:**

The reviewers engaged with the authors, and discussed the paper with the AC.

---

### Decision · Program_Chairs · 2025-01-22

Reject